# Temporospatial induction of homeodomain gene *cut* dictates natural lineage reprogramming

Ke Xu[1], Xiaodan Liu[1], Yuchun Wang[1], Chouin Wong[1], Yan Song[1,2]*

[1]Ministry of Education Key Laboratory of Cell Proliferation and Differentiation, School of Life Sciences, Peking University, Beijing, China; [2]Peking-Tsinghua Center for Life Sciences, Peking University, Beijing, China

**Abstract** Understanding how cellular identity naturally interconverts with high efficiency and temporospatial precision is crucial for regenerative medicine. Here, we revealed a natural midgut-to-renal lineage conversion event during *Drosophila* metamorphosis and identified the evolutionarily-conserved homeodomain protein Cut as a master switch in this process. A steep Wnt/Wingless morphogen gradient intersects with a pulse of steroid hormone ecdysone to induce *cut* expression in a subset of midgut progenitors and reprogram them into renal progenitors. Molecularly, ecdysone-induced temporal factor Broad physically interacts with *cut* enhancer-bound Wnt pathway effector TCF/β-catenin and likely bridges the distant enhancer and promoter region of *cut* through its self-association. Such long-range enhancer-promoter looping could subsequently trigger timely *cut* transcription. Our results therefore led us to propose an unexpected poising-and-bridging mechanism whereby spatial and temporal cues intersect, likely via chromatin looping, to turn on a master transcription factor and dictate efficient and precise lineage reprogramming.
DOI: https://doi.org/10.7554/eLife.33934.001

*For correspondence:
yan.song@pku.edu.cn

**Competing interests:** The authors declare that no competing interests exist.

## Introduction

Classical regenerative strategies are facing challenges, including difficulties associated with the acquisition, delivery and integration of proper cell types into a complex milieu of tissues. In vivo lineage reprogramming, conversion of a highly specialized cell into the desired cell identity, has therefore emerged as an alternative and promising regenerative strategy (*Heinrich et al., 2015*; *Jopling et al., 2011*). However, molecular mechanisms underlying in vivo lineage conversion remained obscure (*Heinrich et al., 2015*; *Jopling et al., 2011*). Due to its high efficiency and temporospatial precision, rare naturally-occurring lineage reprogramming events provide powerful model systems for elucidating the molecular basis of cell plasticity and identity switch.

Here, we revealed a unique naturally-occurring midgut-to-renal lineage reprogramming event at the onset of *Drosophila* metamorphosis. *Drosophila* excretory system, so-called Malpighian tubules, are two pairs of tubules converge through common ureters onto midgut-hindgut junction (*Figure 1—figure supplement 1A*) (*Denholm and Skaer, 2009*; *Dow, 2009*; *Singh et al., 2007*). Each pair of renal tubules can be mainly divided into three segments: ureter, lower tubule and upper tubule (*Figure 1A,B* and *Figure 1—figure supplement 1A*) (*Singh et al., 2007*; *Sözen et al., 1997*). The ureter can be further divided into lower and upper regions (*Figure 1B*). Renal stem cells (RSCs) were found to be dispersed in the adult ureter and lower tubule regions (*Figure 1B*) (*Singh et al., 2007*) but not in the larval renal tubules, raising the question of how the adult RSCs emerge in development. Earlier work (*Takashima et al., 2013*) and our independent observations found that adult RSCs are likely to be derived from progenitors within the midgut region. Midgut progenitors (MPs) and renal progenitors (RPs), although both express Snail-type transcription factor Escargot (Esg), are

**eLife digest** As an embryo develops, an organism transforms from a single cell into an organized collection of different cells, tissues and organs. Regulated by genes and messenger molecules, non-specialized cells known as precursor cells, move, divide and adapt to produce the different cells in the adult body.

However, sometimes already-specialized adult cells can acquire a new role in a process known as lineage reprogramming. Finding ways to artificially induce and control lineage reprogramming could be useful in regenerative medicine. This would allow cells to be reprogrammed to replace those that are lost or damaged. So far, scientists have been unable to develop a clear view of how lineage reprogramming happens naturally.

Here, Xu et al. identified a cell-conversion event in the developing fruit fly. As the fly larva develops into an adult, a group of cells in the midgut reprogramme to become renal cells – the equivalent to human kidney cells. The experiments revealed that a combination of signals from a cell messenger system important for cell specialization (called Wnt) and the hormone that controls molting in insects, activate a gene called cut, which controls the midgut-to-renal lineage reprogramming.

Together, Wnt and the hormone ensure that cut is activated only in a small, specific group of midgut precursor cells at a precise time. The reprogrammed cells then move into the excretory organs, the renal tubes, where they give rise to renal cells. Midgut precursor cells in which cut had been experimentally removed, still traveled into the renal tubes. However, they failed to switch their identity and gave rise to midgut cells instead.

Further examination revealed that both Wnt and the ecdysone hormone are needed to activate the cut gene. This is probably achieved by creating loops in the DNA to bring together the two distantly located key regulatory elements of cut gene expression. If this mechanism can be seen in other contexts it may be possible to adapt it for medical purposes. The ability to reprogramme groups of cells with high specificity could transform medicine. It would make it easier for our bodies to regenerate and repair.

DOI: https://doi.org/10.7554/eLife.33934.002

distinct populations of precursor cells in terms of lineage composition and functionality: midgut progenitors/stem cells undergo asymmetric cell divisions to self-renew and meanwhile differentiate into hormone/peptide-secreting enteroendocrine (EE) cells and nutrient-absorbing enterocytes (ECs) (*Micchelli and Perrimon, 2006*; *Ohlstein and Spradling, 2006*); in contrast, renal progenitors undergo asymmetric, self-renewing divisions to give rise to principal cells that mediate organic cation and solute transport (*Singh et al., 2007*). Intriguingly, we observed that, during metamorphosis, a small subset of Esg$^+$ progenitors appeared to migrate away from the midgut and onto the renal tubules (*Figure 1C–E*), where they terminally differentiated into new Cut$^+$ principal cells (arrowheads in *Figure 1D*), replacing the old Cut$^-$ principal cells in the lower ureter region (arrowheads in *Figure 1C*) (*Takashima et al., 2013*). However, it remains enigmatic when, where and how the pool of Esg$^+$ midgut progenitors is selected and converted into renal identity during metamorphosis.

To ascertain this midgut-to-renal lineage conversion event and, more importantly, to probe its underlying regulatory mechanisms and molecules, we carried out a genome-wide unbiased RNAi-based genetic screen (Xu et al., unpublished) and identified the homeodomain protein Cut as a master switch dictating this lineage reprogramming event. Cut is an evolutionarily-conserved homeodomain-containing transcription factor that has been shown to regulate various developmental events in *Drosophila* and mammals, including sensory organ identity specification and dendritic morphogenesis in peripheral nervous system, dorsal-ventral boundary formation in the fly wings, projection neuron dendritic targeting, as well as patterning and growth during fly airway remodeling (*Becam et al., 2011*; *Blochlinger et al., 1988*; *Bodmer et al., 1987*; *Cubelos et al., 2010*; *Grueber et al., 2003*; *Komiyama and Luo, 2007*; *Ludlow et al., 1996*; *Pitsouli and Perrimon, 2013*; *Rodríguez-Tornos et al., 2016*).

Here, we show that a steep Wnt/Wingless (Wg) morphogen gradient (*Clevers and Nusse, 2012*; *Loh et al., 2016*) at the midgut-hindgut boundary intersects with a pulse of the steroid hormone

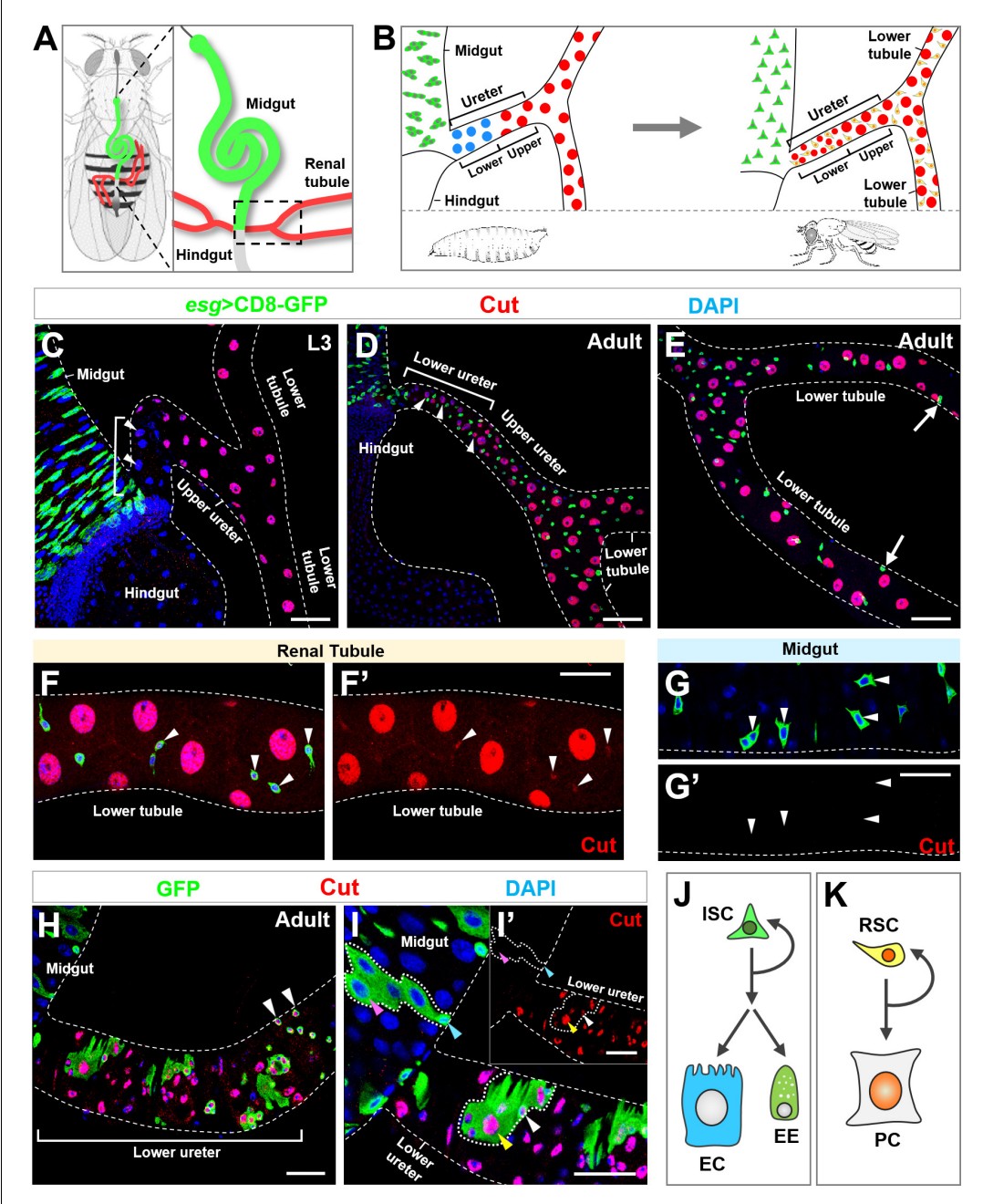

**Figure 1.** Homeodomain transcription factor Cut is specifically expressed in *Drosophila* adult renal stem cells. (A) A schematic diagram of two pairs of renal tubules (red) that converge at ureters and connect to the digestive tract at the midgut (green)-hindgut (grey) boundary of an adult fly (A). The area encircled by dashed line in (A) is magnified and shown in (B). (B) Close-up schematics of larval (left) and adult (right) intestine and renal tubules. Note that each pair of renal tubules merges together at the ureter that is further divided into lower and upper regions. Adult renal stem cells (RSC; yellow) are present in adult but not larval renal tubules. The large principal cells (PC) in lower ureter (blue) during larval stage are replaced with intermediate sized new principal cells (red) during adult stage. (C) Progenitors marked by *esg^{ts}*>CD8-GFP (green) reside in the third instar larval (L3) midgut. White bracket indicates the lower ureter region containing Cut⁻ principal cells (blue). (D–E) During adult stage, Esg⁺ stem cells are present in both midgut and renal tubules. White bracket indicates lower ureter containing Cut⁺ Esg⁻ principal cells (D). Arrows indicate furthest detectable RSCs on renal tubules (E). (F–G') Cut is expressed in adult renal stem cells (arrowheads in F, F') but not intestinal stem cells (ISC; arrowheads in G, G'). (F',G') show single-channel images for Cut immunostaining. (H) MARCM clones induced at early larval stages and examined at early adult stage. Clones containing multiple cells appeared only at lower ureter (bracket), whereas single-cell clones containing Cut⁺ progenitors (arrowheads) appeared beyond that region. (I, I') MARCM clones induced at lower ureter contain Cut⁺ small diploid RSCs (white arrowhead) and Cut⁺ large polyploid principal cells (yellow

*Figure 1 continued on next page*

*Figure 1 continued*

arrowhead). The inset shows single-channel image for Cut immunostaining (I'). Neither ISCs (cyan arrowhead) nor polyploid enterocytes (ECs; pink arrowhead) expressed Cut (I'). (J, K) Distinct composition of ISC (J) and RSC (K) lineages. EE: enteroendocrine. Scale bar: 50 µm (C–E); 25 µm (F–I').

DOI: https://doi.org/10.7554/eLife.33934.003

The following figure supplement is available for figure 1:

**Figure supplement 1.** Identification of Cut as a specific adult renal stem cell marker.

DOI: https://doi.org/10.7554/eLife.33934.004

ecdysone at the onset of metamorphosis (*Yamanaka et al., 2013*) to induce *cut* expression in a subset of midgut progenitors and reprogram them into renal progenitors. Mechanistically, the Wg morphogen gradient, through its pathway effector TCF/β-catenin, determines the pool of future renal progenitors, presumably by poising a distal *cut* enhancer for timely activation. On the other hand, the hormone ecdysone-induced BTB-Zinc finger protein Broad determines the timing of lineage conversion by physically interacting with enhancer-bound TCF/β-catenin complex and likely bridging the distal enhancer and promoter region of *cut* through its self-association. Such long-range enhancer-promoter looping could subsequently trigger timely *cut* transcription. Thus, integration of spatial and temporal cues by a master cell identity switch, likely through a chromatin looping mechanism, orchestrates natural lineage reprogramming with temporal and spatial precision.

## Results

### Adult renal stem cells specifically express homeodomain protein Cut

To identify key regulators governing midgut-to-renal lineage conversion, we carried out a genome-wide RNAi-based screen. We used temperature-sensitive, midgut and renal progenitor-specific *esg*-Gal4, UAS-CD8-GFP, tub-Gal80$^{ts}$ system to drive RNAi expression, transferred animals from permissive temperature (18°C) to restrictive temperature (29°C) at mid third instar larval stage, approximately 48 hr before the lineage reprogramming event occurs, and analyzed the renal phenotypes at early adult stages. Intriguingly, we found that midgut progenitor-specific knockdown of the transcription factor Cut resulted in lack of the entire lower ureter region (brackets in *Figure 1—figure supplement 1B*) and appearance of extra Esg$^-$ diploid cells along the renal tubules (arrowheads in *Figure 1—figure supplement 1B*).

Such striking renal tubule phenotypes of *cut-RNAi* prompted us to carefully investigate the expression pattern of Cut in adult intestine and renal tubules. Cut has previously been found to be highly expressed in two types of post-mitotic, polyploid cells in the *Drosophila* digestive-excretive system: the acid-secreting copper cells in the middle midgut region (*Strand and Micchelli, 2011*) and the fluid-balancing principal cells within renal tubules (*Singh et al., 2007*; *Singh et al., 2011*). Surprisingly, we detected moderate Cut expression in Esg$^+$ diploid cells along adult renal tubules (arrowheads in *Figure 1F,F'*). Knockdown or overexpression of Cut within adult renal stem cells, by *esg*-Gal4, diminished or elevated the Cut expression levels respectively (*Figure 1—figure supplement 1C*), indicating that such Cut expression is specific. In contrast, Cut was not detectable in adult intestinal stem cells or enteroblasts (EBs) in midgut (arrowheads in *Figure 1G,G'*). To further confirm the specific expression of Cut in renal stem cell lineages, we induced wild-type MARCM (mosaic analysis with repressible cell marker) clones (*Lee and Luo, 2001*) at larval stages and analyzed Cut expression pattern of clones at early adult stage. Using the MARCM system, the homozygous clones are produced upon mitotic recombination and are positively labeled by GFP. MARCM clones induced at the lower ureter region (bracket in *Figure 1H*) contained both diploid cells expressing moderate levels of Cut (white arrowheads in *Figure 1I,I'*) and polyploid cells expressing high levels of Cut (yellow arrowheads in *Figure 1I,I'*), indicating that new Cut$^+$ principal cells are derived from Cut$^+$ renal progenitors (*Figure 1B,D,I*). By contrast, MARCM clones in the midgut region, composed of diploid ISCs (cyan arrowheads in *Figure 1I*), EBs, EEs and polyploid ECs (pink arrowheads in *Figure 1I*), did not exhibit Cut expression. Therefore, distinct from Cut$^-$ intestinal stem cells (ISCs) that differentiate into Cut$^-$ EEs and ECs in the midgut (*Figure 1J*), Cut$^+$ renal stem cells (RSCs) differentiate into Cut$^+$ principal cells (PCs) on the renal tubules (*Figure 1K*). Taken together, we identified Cut as a specific marker that distinguishes adult RSCs from ISCs.

## Adult RSCs are originated from Cut[+] progenitors within midgut

Utilizing Cut as a cell identity marker, we studied the emergence, proliferation, migration and differentiation of RPs during metamorphosis. Cut was undetectable in any Esg[+] progenitors in third instar larval (L3) midguts (*Figure 2A*). At the onset of metamorphosis (0 hr after puparium formation (APF)), a few clusters of Esg[+] progenitors in closest proximity to the midgut-hindgut boundary started to exhibit Cut expression (white arrowheads in *Figure 2B*). At 0.5 hr APF, 20–30 Esg[+] progenitors in 5–6 clusters specifically expressed Cut (yellow bracket in *Figure 2C*). By 1 hr APF, midgut progenitor islands merged together and the Cut[+] progenitors aligned along the midgut-hindgut border in 1–3 rows (yellow bracket in *Figure 2D*). Starting at 3 hr APF, Esg[+] Cut[+] progenitors migrated across the midgut border and entered the renal tubules (yellow bracket in *Figure 2E*). Cut[-] principal cells in the lower ureter region were engulfed and expelled into the pupal midgut (pink arrowheads in *Figure 2E*) (*González-Morales et al., 2015*). Cut[+] progenitors in turn occupied the lower ureter region (*Figure 2F*) and differentiated into Esg[-] Cut[+] new principal cells (cyan arrowheads in *Figure 2G*). The dynamic changes in the number of Cut[+] progenitors and the emergence, proliferation, migration and differentiation of RPs during metamorphosis are schematically presented in *Figure 2H and I* respectively.

At the onset of metamorphosis, Cut[+] progenitors expressed midgut progenitor marker Delta (Dl) (arrowheads in *Figure 2—figure supplement 1A,B*), indicating that Cut[+] progenitors were transited from MP to RP characteristics. Interestingly, immediately after the Cut[+] progenitors migrated onto renal tubules, they started to turn off Dl expression (arrowheads in *Figure 2—figure supplement 1C*). By 9 hr APF, Cut[+] progenitors along the renal tubules completely shut down Dl expression (arrowheads in *Figure 2—figure supplement 1D*). Therefore, Cut[+] progenitors are progressively reprogrammed from MPs to RPs.

## RPs displayed MP characteristics upon Cut depletion

The highly specific and restrictive induction of Cut expression in RPs right before their migration onto renal tubules prompted us to investigate whether Cut plays a critical role in the identity switch and/or migration of future RPs. Upon *cut* knockdown by *esg*-Gal4, the migration of the small subset of Esg[+] progenitors onto renal tubules remained relatively normal (*Figure 3A–D'*), indicating that Cut is dispensable for progenitor mobility. To investigate whether Cut is important for the identity switch of Esg[+] progenitors, we performed lineage-tracing experiments based on the G-TRACE (the Gal4 Technique for Real-time and Clonal Expression) system (*Evans et al., 2009*). We used temperature-sensitive *esg*-Gal4, UAS-CD8-GFP, tub-Gal80[ts] system to drive FLP (flippase) expression, transferred animals from permissive temperature (18°C) to restrictive temperature (29°C) at late third instar larval stage and analyzed the renal phenotypes at early adult stages (*Figure 3A,B*). Upon *esg*-Gal4-driven expression of FLP recombinase, a transcriptional stop cassette flanked by FRT sites is excised, resulting in lacZ expression in Esg[+] progenitors and all their subsequent daughter cells (lineage expression; *Figure 3A*). Meanwhile, GFP reveals the real-time expression of *esg*-Gal4 (*Figure 3A*). In wild type renal tubules, Esg[-] lacZ[+] cells derived from Esg[+] lacZ[+] RPs (white arrowheads in *Figure 3C,C'*) were found in the lower ureter region (brackets in *Figure 3C,C'*). These Esg[-] lacZ[+] cells were polyploid, new Cut[+] principal cells (white arrowheads in *Figure 3C,C'*), in accordance with our earlier notion that Cut[-] principal cells are replaced with RP-derived new Cut[+] principal cells during metamorphosis. *cut* knockdown by *esg*-GAL4 caused the majority of flies to die at late pupal or early adult stages (90.3%, n = 567). The adult escaper flies lacked the whole lower ureter region (*Figure 3D,D'*), hinting that maldevelopment of the renal tubules contributed to the lethality of the animals. In contrast to the wild type control (*Figure 3C,E*), Esg[-] lacZ[+] cells derived from *cut*-RNAi RPs were dispersed throughout the ureter and lower tubule regions (white arrowheads in *Figure 3D'*). Strikingly, these Esg[-] lacZ[+] cells were diploid, Cut[-] (*Figure 3D,D'*), and expressed Prospero (Pros) (*Figure 3F,F'*), a typical marker for EE cells. These results strongly suggested that, upon *cut* depletion, Esg[+] progenitors migrate normally onto the renal tubules, yet fail to switch into renal identity and differentiate into midgut cells along renal tubules.

Importantly, upon *cut* depletion, Esg[+] progenitors along renal tubules behaved essentially the same as MPs at pupal stages (*Guo and Ohlstein, 2015*): (1) they produced ectopic Pros[+] diploid cells along renal tubules at the same developmental stages as wild type MPs giving rise to EEs in midgut (arrowheads in *Figure 3—figure supplement 1A,B*); (2) The ectopic Pros[+] cells derived from

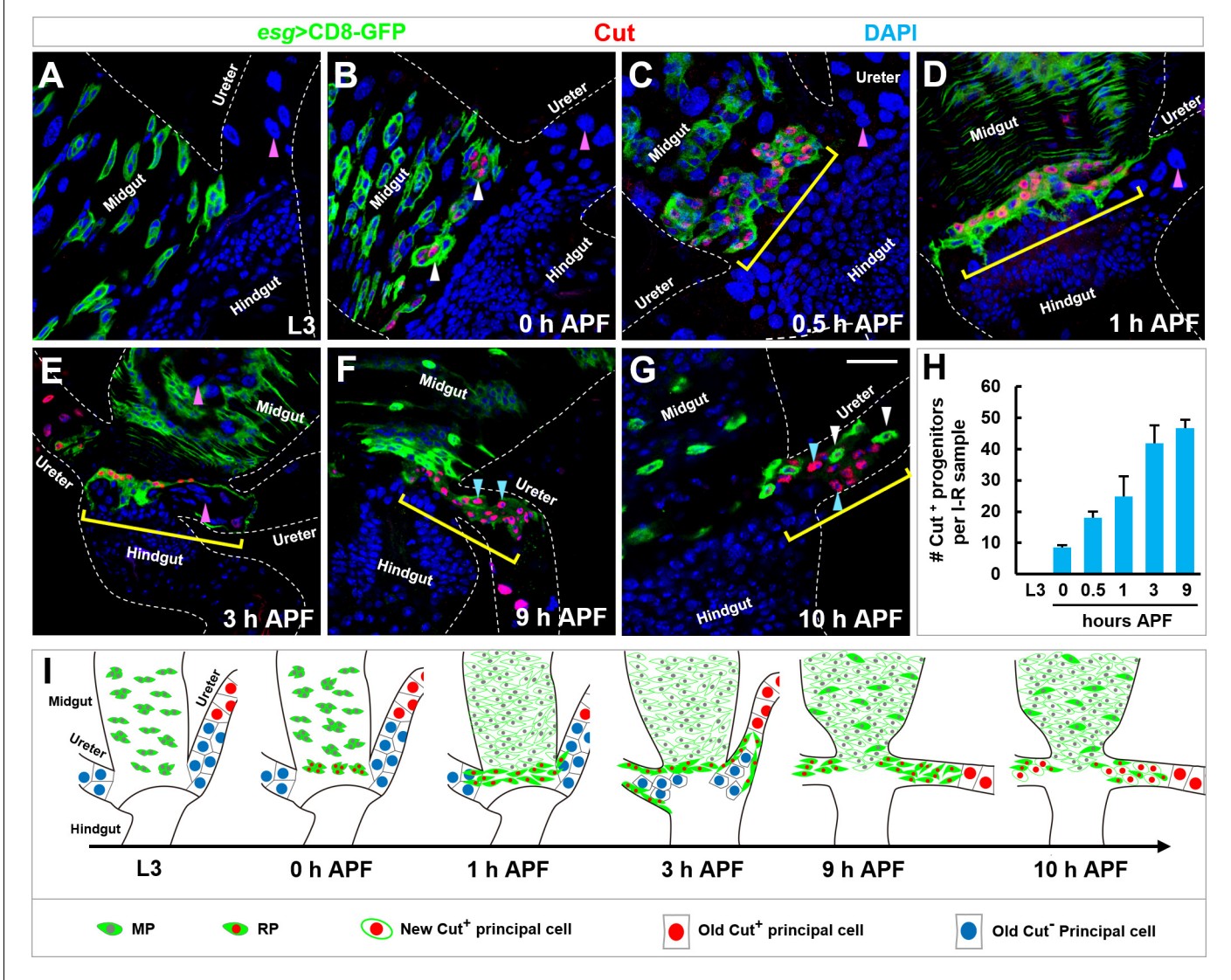

**Figure 2.** Emergence, migration and differentiation of Cut[+] renal progenitors during metamorphosis. (A–G) Midgut-hindgut boundary and lower ureter region at the stages of (A) third instar larvae (L3), (B) 0 hr, (C) 0.5 hr, (D) 1 hr, (E) 3 hr, (F) 9 hr or (G) 10 hr after puparium formation (APF) stained with Cut (red). Note that Cut[+] progenitors are marked with white arrowheads (B) or yellow brackets (C–G). Also note that Cut[-] principal cells in the lower ureter (pink arrowheads in A–D) are engulfed and deposited to the intestinal lumen at 3 hr APF (pink arrowheads in E). Upon differentiation into Cut[+] principal cells (cyan arrowheads in F and G), Esg[+] progenitors exhibit reduced levels of Cut expression (white arrowheads in G). (H) Quantification of the number of Cut[+] progenitors per intestine-renal tubule (I-R) sample at different developmental stages (n = 7–13). (I) Schematic representations of emergence, migration and differentiation of Esg[+] Cut[+] RPs. Scale bar, 25 μm.

DOI: https://doi.org/10.7554/eLife.33934.005

The following source data and figure supplement are available for figure 2:

**Source data 1.** Input data for bar graph *Figure 2H*.

DOI: https://doi.org/10.7554/eLife.33934.007

**Figure supplement 1.** Cut[+] renal progenitors are converted from Dl[+] midgut progenitors.

DOI: https://doi.org/10.7554/eLife.33934.006

them expressed additional EE markers such as exocytosis regulator Rab3 (*Dutta et al., 2015*; *Patel et al., 2015*), synaptic protein Bruchpilot (Brp) (*Zeng et al., 2013*), and secretory neuropeptide hormone Allatostatin A (Ast A) (*Beehler-Evans and Micchelli, 2015*) (yellow arrowheads in *Figure 3—figure supplement 1C–E*), indicating that they are bona fide EE cells; (3) they underwent asymmetric cell divisions, resulting in unidirectional Notch activation in progenitors (open

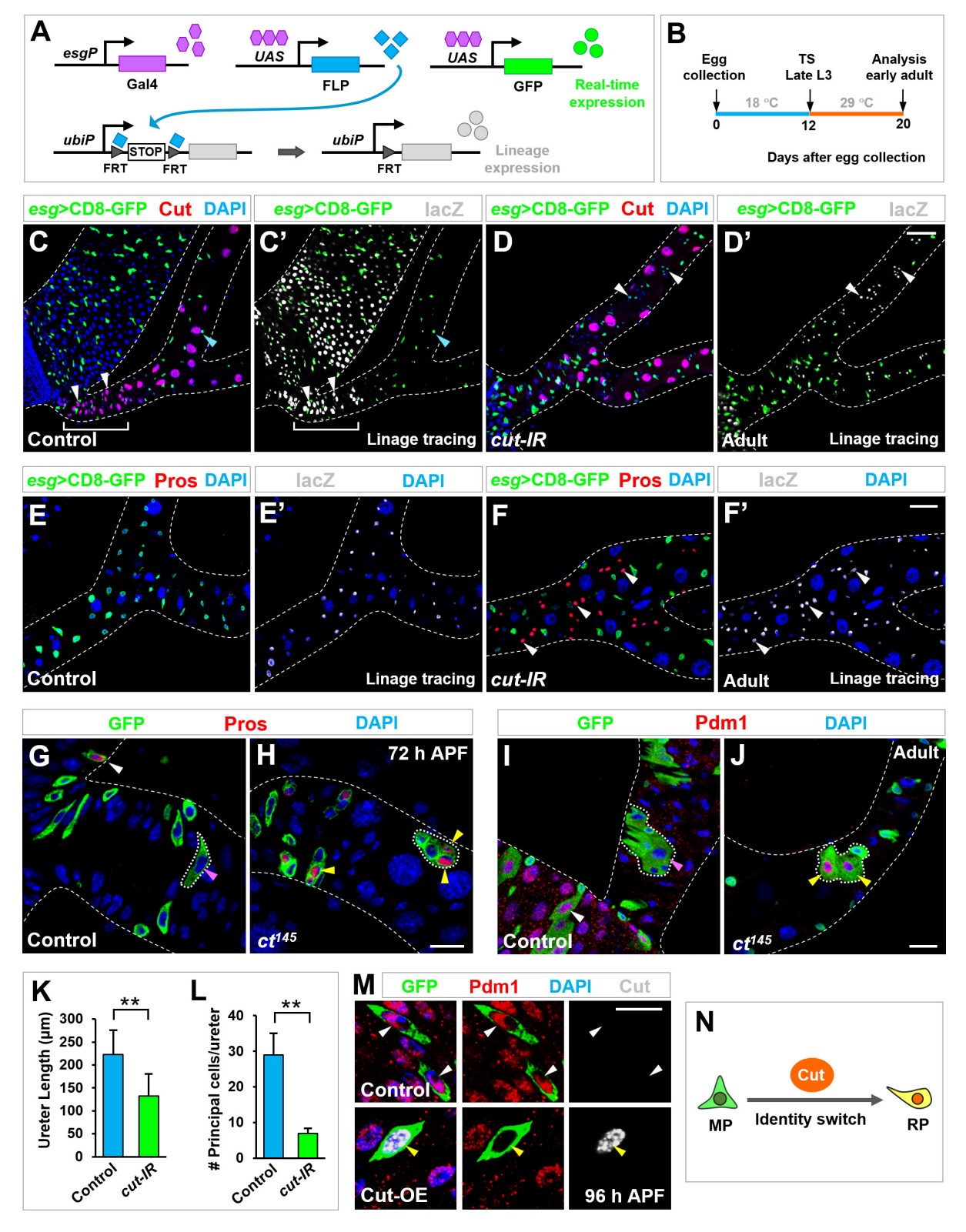

**Figure 3.** Cut dictates midgut-to-renal progenitor identity switch. (**A–B**) Genetic schema (**A**) and experimental timeline (**B**) of the lineage-tracing experiment. (**C–F'**) Lineage-tracing analysis of Esg+ progenitors in ureter and lower tubule regions. Wild type RPs in lower ureter (white brackets in **C**, **C'**) differentiate into Cut+ polyploid principal cells (white arrowheads in **C**, **C'**). Upon *cut* depletion, Esg+ progenitors in ureter and lower tubule regions give rise to Cut- Pros+ diploid cells (white arrowheads in **D'**, **F**, **F'**). Note that the *cut-RNAi* renal tubules lack the entire lower ureter region (**D**, **D'**). (**G–J**)
*Figure 3 continued on next page*

*Figure 3 continued*

Wild type control MARCM clones residing in the midgut but not ureter region (pink arrowheads in **G** and **I**) contain Pros⁺ EE cells (white arrowhead in **G**) and Pdm1⁺ ECs (white arrowhead in **I**). In sharp contrast, *cut* mutant clones at ureter region contain Pros⁺ EEs (yellow arrowheads in **H**) or Pdm1⁺ ECs (yellow arrowheads in **J**). (**K–L**) Quantification of the length of ureter (**K**; **p<0.001 (n = 11)) and the total number of principal cells (**L**; **p<0.0001 (n = 7–9)) in wild type or *esg^ts* >*cut*-IR adult animals. (**M**) Control or Cut-overexpressing (Cut-OE) MARCM clones were induced at third instar larval stages, dissected at late-pupal stage and stained with Pdm1 (red) and Cut (gray). Note that 100% polyploid cells in WT control clones are Pdm1⁺ Cut⁻ (n = 69). By contrast, 91% polyploid cells in Cut-OE clones are Pdm1⁻ Cut⁺, with the rest 9% expressing low levels of Pdm1 and Cut (n = 66). (**N**) Schematic depiction of Cut as a MP-to-RP identity switch. Scale bars, 50 μm (**C–D′**), 25 μm (**E–F′, I, J**) and 10 μm (**G, H, M**).

DOI: https://doi.org/10.7554/eLife.33934.008

The following source data and figure supplements are available for figure 3:

**Source data 1.** Input data for bar graph *Figure 3K,L*.
DOI: https://doi.org/10.7554/eLife.33934.011
**Figure supplement 1.** Cut-depleted Esg⁺ progenitors in renal tubules behave exactly like midgut progenitors.
DOI: https://doi.org/10.7554/eLife.33934.009
**Figure supplement 2.** Notch signaling modulates EE and EC differentiation on renal tubules.
DOI: https://doi.org/10.7554/eLife.33934.010

arrowheads in *Figure 3—figure supplement 2A–D*) and asymmetric localization of Pros to the basal context (*Figure 3—figure supplement 2E*); and (4) their cell fate decisions were tightly regulated by the strength of Notch signaling (*Figure 3—figure supplement 2F–H*).

To further confirm the notion as proposed above, we carried out MARCM clonal analysis. GFP-marked clones were induced at second instar stage and analyzed at mid pupal (*Figure 3G,H*) or early adult stages (*Figure 3I,J*). Pros⁺ EEs or Pdm1⁺ polyploid ECs were present in wild type MARCM clones in the midgut (white arrowheads in *Figure 3G,I*) but not renal tubule regions (pink arrowheads in *Figure 3G,I*). In sharp contrast, both EEs and ECs were found within *cut* mutant MARCM clones in the renal tubule regions (yellow arrowheads in *Figure 3H,J*). Moreover, as a consequence of *cut* depletion from Esg⁺ progenitors, both the total length of the ureter region (*Figure 3K*) and the number of principal cells in ureter (*Figure 3L*) were sharply reduced. Taken together, our results clearly demonstrated that Cut is necessary for MP-to-RP identity switch.

To test whether Cut is sufficient in dictating midgut-to-renal lineage conversion, we overexpressed Cut in MPs and analyzed their lineage progression using MARCM clonal analysis. In wild type MARCM clones derived from single MPs, GFP⁺ polyploid cells were Pdm1⁺ ECs (white arrowheads in *Figure 3M*). In contrast, MARCM clones derived from Cut-overexpressing MPs mainly contained Pdm1⁻ Cut⁺ polyploid cells (yellow arrowheads in *Figure 3M*), strongly suggesting that MPs were converted into RPs that in turn differentiated into principal cells in midgut region. Collectively, our observations reinforced the idea that Cut acts as a master switch in dictating natural midgut-to-renal progenitor identity conversion (*Figure 3N*).

## Wnt/Wg morphogen acts as a spatial cue in inducing Cut expression in MPs

We next sought to identify the signaling cues that induced Cut expression in the specific subsets of MPs. The spatially restricted distribution of Cut⁺ MPs hinted that Cut expression might be induced by a morphogen gradient emanating from proximal cells. After examining the distribution pattern of various signaling molecules or receptors at the onset of metamorphosis, we found that, in accordance with previous observations (*Fox and Spradling, 2009*; *Takashima et al., 2008*; *Tian et al., 2016*), the Wnt/Wingless (Wg) ligand (*Clevers and Nusse, 2012*; *Loh et al., 2016*) was expressed in a narrow zone of 2–3 rows of cells right at the midgut-hindgut boundary, starting from early larval stages (*Figure 4A,B* and *Figure 4—figure supplement 1A,B*). At the onset of metamorphosis, 2–3 clusters of progenitors in closest proximity to the stripe of Wg-producing cells started to turn on Cut expression (arrowheads in *Figure 4C* and *Figure 4—figure supplement 1C*). Around 0.5 hr APF, Cut was expressed in 5–6 progenitor islands adjacent to the Wg-secreting cells (arrowheads in *Figure 4D*). Some Cut⁺ progenitor islands were not in direct contact with the Wg-producing band (arrowheads in *Figure 4D*), suggesting the existence of Wg morphogen gradient. At 1 hr APF, peripheral cells surrounding progenitor islands (*Mathur et al., 2010*) partially opened up, allowing Cut⁺ progenitor islands to merge with each other (*Figure 4E*). A clear boundary between Cut⁺ and

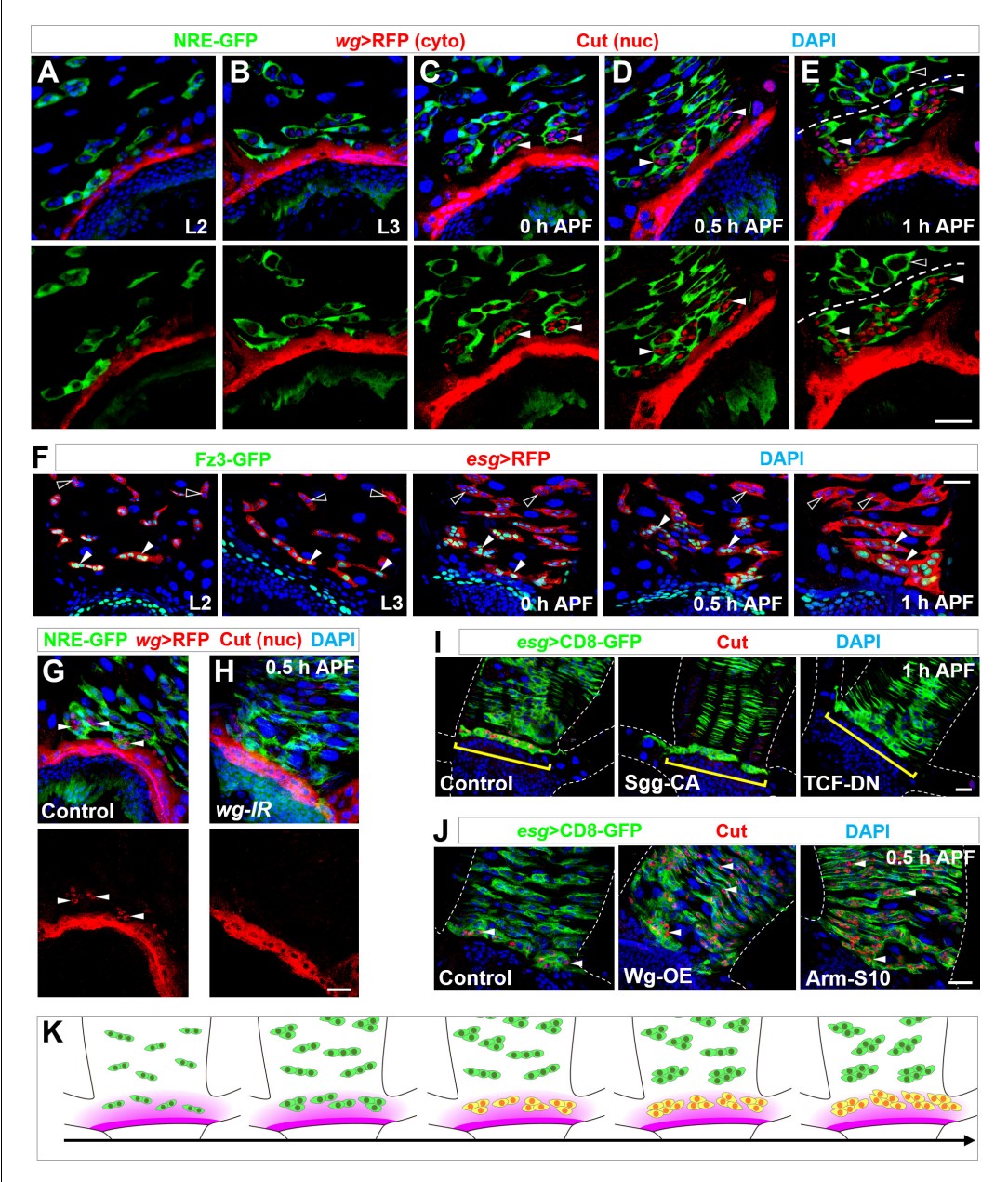

**Figure 4.** Wg morphogen acts as a spatial cue selecting the pool of future RPs. (A–E) Wg ligand expression, as labeled by *wg*-Gal4 >CD8-RFP, is highly restricted to a narrow band of cells proximal to the midgut-hindgut boundary at the stages of second instar (A) or third instar larva (B), or 0 hr (C), 0.5 hr (D) or 1 hr (E) APF. NRE-GFP marks peripheral cells encasing midgut progenitor islands. Note that Cut+ progenitors (red, nuclear signal; indicated by arrowheads) are in close proximity to the Wg-producing band. Cyto: cytoplasmic; nuc: nuclear. (F) Expression pattern of Wg signaling reporter Fz3-GFP at different developmental stages. Closed and open arrowheads indicate Fz3-GFP+ and Fz3-GFP- progenitors respectively. (G,H) *wg*-Gal4>*white*-RNAi or *wg*-Gal4>*wg*-RNAi midgut-hindgut boundary region stained with Cut (nuclear signal; arrowheads in G). Note that *white-RNAi* serves here as a negative control. (I) Upon overexpression of Sgg-CA or TCF-DN, Esg+ progenitors in closest proximity to the band of *wg*-producing cells (brackets) fail to express Cut. (J) Upon overexpression of Wg or Arm-S10, Esg+ progenitors far away from the midgut-hindgut boundary ectopically turn on Cut expression (arrowheads). (K) Schematic drawings of the progressive MP (green)-RP (yellow) identity switch in the range of Wg morphogen gradient (purple) along the developmental axis. Scale bars, 25 μm.

DOI: https://doi.org/10.7554/eLife.33934.012

The following figure supplement is available for figure 4:

**Figure supplement 1.** Wg signaling is both necessary and sufficient for inducing *cut* expression in Esg+ progenitors.

DOI: https://doi.org/10.7554/eLife.33934.013

Cut⁻ progenitors could be delineated (dashed lines in *Figure 4E* and *Figure 4—figure supplement 1D*), suggesting the spatial precision of the inductive cue. Wg signaling activity, as faithfully reflected by the Fz3-GFP reporter (*Sivasankaran et al., 2000*; *Tian et al., 2016*), was only detectable in a few rows of MPs proximal to Wg-secreting cells (closed arrowheads in *Figure 4F*), further supporting Wg signaling as a spatial cue in selecting future RPs.

If Wg signaling provides an inductive cue, we reasoned that downregulation of Wg signaling in future RPs should abolish Cut expression. Indeed, specific knockdown of Wg within Wg-producing cells at midgut-hindgut boundary, by *wg*-Gal4 (*Alexandre et al., 2014*), resulted in diminished Cut expression in Esg⁺ progenitors (arrowheads in *Figure 4G,H*). Furthermore, inhibition of Wg signaling through overexpression of either a dominant-negative form of the Wnt pathway effector TCF (ΔN-TCF) (*van de Wetering et al., 1997*) or a constitutively-active form of the Wnt pathway inhibitor GSK3β/Shaggy (Sgg-CA) (*Bourouis, 2002*) in Esg⁺ progenitors completely abolished Cut induction (brackets in *Figure 4I*). Consistently, MARCM clonal analysis revealed that, upon depletion of the Wnt/Wg pathway positive component Disheveled (Dsh), ectopic Pros⁺ EEs appeared in renal tubes (yellow arrowheads in *Figure 4—figure supplement 1E*). Therefore, Wg signaling is essential for the MP-to-RP identity switch.

We next investigated why Cut was expressed in only a small subset of Esg⁺ progenitors by probing the competence of MPs to respond to Wg signaling. Upon overexpression of Wg ligand in all MPs, high levels of Cut expression were detected in all Esg⁺ progenitors dispersed throughout the midgut (arrowheads in *Figure 4J*). As a consequence, elevated number of Cut⁺ progenitors migrated onto renal tubules (*Figure 4—figure supplement 1F,G*). In accordance, activation of Wg signaling in all MPs through overexpression of a constitutively-active form of the Wnt pathway effector β-catenin/Armadillo (Arm-S10) (*Pai et al., 1997*) exhibited similar effects as Wg overexpression (arrowheads in *Figure 4J*), demonstrating that all MPs are competent to respond to Wg signaling. Taken together, our data strongly support the notion that a steep Wg morphogen gradient provides a spatial cue to precisely select the pool of future RPs during metamorphosis (*Figure 4K*).

To find out to what extent the spread of Wg is required for Cut induction, we employed a membrane-tethered form of Wg, *wg*(KO; NRT-Wg) (*Alexandre et al., 2014*). While membrane-tethered Wg sufficed to control fly patterning and growth (*Alexandre et al., 2014*), it failed to induce Cut expression in future RPs (*Figure 4—figure supplement 1H*), indicating that the spread of Wg is necessary for activating Wg signaling in these progenitors.

## Ecdysone hormone as a temporal cue in midgut-to-renal lineage conversion

The subset of MPs in close proximity to Wg-producing cells do not turn on Cut expression until the onset of metamorphosis, raising the question of how Cut induction is temporally controlled. Since the expression levels of *wg*-Gal4, *wg*-lacZ or Wg signaling reporter Fz3-GFP at the midgut-hindgut boundary remained relatively constant from second instar larval to early pupal stages (*Figure 4A–F* and *Figure 4—figure supplement 1A–D*), temporal cue(s) other than Wg ligand triggers Cut expression in future RPs at the onset of metamorphosis.

The induction of Cut coincided with the pulse of the steroid hormone ecdysone released from the ring glands (*Ou and King-Jones, 2013*; *Yamanaka et al., 2013*), hinting that the ecdysone hormone may serve as a temporal cue (*Praggastis and Thummel, 2017*; *Uyehara et al., 2017*). Indeed, Cut induction was abolished upon Esg⁺ progenitor-specific expression of a dominant-negative form of the Ecdysone receptor (EcR-DN) (*Brown et al., 2006*) (*Figure 5A*). Since EcR was widely expressed in all cell types in the midgut at different developmental stages (*Figure 5—figure supplement 1A,B*), we considered the possibility that the strong pulse of ecdysone at metamorphosis was translated into a temporal patterning of early response genes downstream of EcR. Supporting this notion, Esg⁺ progenitor-specific depletion of the Broad complex (Br-C) (*Figure 5A*), a crucial early response gene of the ecdysone signaling (*Fletcher and Thummel, 1995*; *Karim et al., 1993*), phenocopied the effects of EcR-DN. In comparison, Cut expression was normally turned on in future RPs upon downregulation of E74 and E75, the other two well-characterized ecdysone early response genes (*Figure 5—figure supplement 1C*). These observations indicated that the steroid hormone ecdysone executed its control on Esg⁺ progenitor identity through specific downstream effector Br-C. Strongly supporting this idea, the temporally dynamic expression pattern of Br-C coincides with the strong pulse of ecdysone during metamorphosis: Br-C protein expression was barely detectable

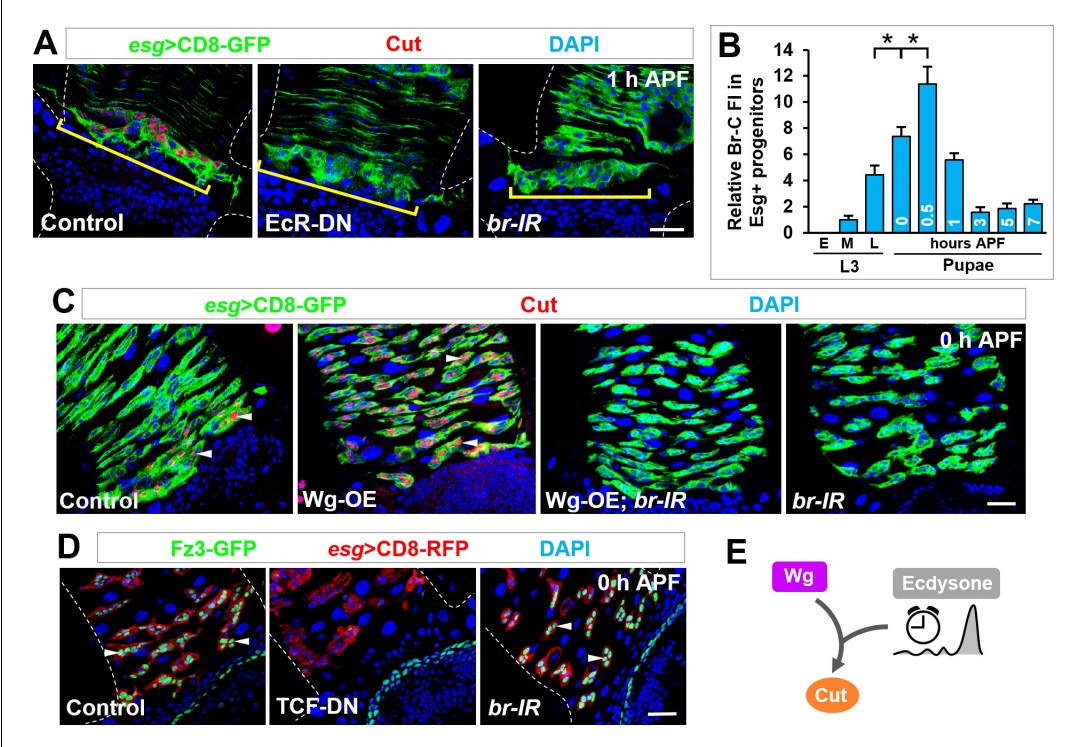

**Figure 5.** Ecdysone hormone acts as a temporal cue and synergizes with Wg signaling in *cut* induction. (**A**) Cut expression in Esg⁺ progenitors (yellow brackets) is diminished upon expression of EcR-DN or *br-RNAi*. (**B**) Quantification of the relative Br-C fluorescent intensity (FI) in Esg⁺ progenitors at different developmental stages. *p<0.0001 (n = 14–16). Note that the Br-C expression levels here are assessed with Br-core antibody that detects all Br isoforms. (**C**) Wg overexpression-induced ectopic expression of Cut in Esg⁺ MPs (arrowheads) is completely abolished upon *br* knockdown. (**D**) Expression pattern of Wg signaling reporter Fz3-GFP of indicated genotypes at 0 hr APF. Note that Fz3-GFP expression in Esg⁺ progenitors is diminished upon TCF-DN but not *br-RNAi* expression. (**E**) Wg and ecdysone signaling pathways converge on *cut* induction.

DOI: https://doi.org/10.7554/eLife.33934.014

The following source data and figure supplement are available for figure 5:

**Source data 1.** Input data for bar graph *Figure 5B*.
DOI: https://doi.org/10.7554/eLife.33934.016

**Figure supplement 1.** Dynamic expression of Broad, in comparison to EcR, and its epistatic interaction with Armadillo in inducing *cut* expression.
DOI: https://doi.org/10.7554/eLife.33934.015

in early third instar larvae, progressively increased starting mid third instar larval stage and peaked at the onset of metamorphosis, followed by a gradual decline (*Figure 5B* and *Figure 5—figure supplement 1D–H*). Together, our results revealed an ecdysone-EcR-Br regulatory axis, which induces Cut expression and dictates progenitor identity switch with temporal precision.

## Cut induction in RPs requires a gene-specific integration of spatial and temporal cues

The above findings identified the Wnt/Wg ligand and the steroid hormone ecdysone as spatial and temporal cues respectively in turning on the master identity switch Cut. This leaves us with the important question of how the spatial and temporal signaling is integrated at molecular level. To address this question, we first carried out epistatic analysis. Depletion of Br-C abolished ectopic Cut expression induced by Wg or Arm-S10 overexpression in MPs (*Figure 5C* and *Figure 5—figure supplement 1I*), demonstrating that Br-C acts downstream of or in parallel with the Wg pathway transcription activation complex TCF/Arm in *cut* induction. Furthermore, Wg pathway activity, as indicated by the Fz3-GFP reporter, was highly responsive to a reduction in Wg signaling, but remained unaltered upon downregulation of ecdysone signaling (*Figure 5D*). This rules out the possibility of a general modulation of the Wg signaling output by the ecdysone pathway. Therefore, it is

likely that Wg and ecdysone signaling converge on the control of *cut* expression in a gene-specific manner (*Figure 5E*).

## Broad forms a transcription activation complex with TCF/Arm in *cut* induction

Given that Br itself is a BTB-ZF transcription factor, it may physically associate with TCF/Arm to form a transcription activation complex and synergistically trigger *cut* transcription. To test this idea, we first carried out coimmunoprecipitation (coIP) assays. The *Br-C* gene locus encodes four distinct splicing isoforms, Br-Z1, Br-Z2, Br-Z3 and Br-Z4, which share a common N-terminal core domain but have distinct C-terminal zinc-finger domains (*Figure 6A*) (*Mugat et al., 2000*). Indeed, we found that Arm could be specifically coimmunoprecipitated with Br-C isoforms from HEK293T cell extracts and exhibited relatively strong binding affinity to Br-TNT-Z1, Br-Z2 and Br-Z4 (*Figure 6B*).

The results described above prompted us to probe the functional significance of the physical interaction between Br-C and TCF/Arm. While overexpression of Br-Z1, Br-Z2 or TCF alone was barely able to precociously induce *cut* transcription at 9 hr before puparium formation (BPF) (induction rate of 0, 36 and 0% respectively; n = 11–12; arrowheads in *Figure 6C*), coexpression of Br-Z1 or Br-Z2 with TCF dramatically enhanced Br activity in premature induction of *cut* expression (induction rate of 70 and 100% respectively; n = 9–17; arrowheads in *Figure 6C*), providing compelling evidence for a combinatorial regulation of *cut* transcription by Br and Arm/TCF. Simultaneous overexpression of TCF and either Br-Z3 or Br-Z4 failed to induce premature *cut* expression, highlighting the importance of the isoform-specific C-terminal domain for Br functionality.

We next examined the temporal expression pattern of Br-Z1 and Br-Z2. While Br-Z2, as stained by our newly-raised antibody (*Figure 6—figure supplement 1A*), exhibited prominent and specific expression in Esg$^+$ progenitors at 0 hr APF, Br-Z1 was undetectable in the midgut region (*Figure 6D* and *Figure 6—figure supplement 1B*). Significantly, the temporal expression pattern of Br-Z2 in Esg$^+$ progenitors fully recapitulated that of Br-C (*Figure 6E*). Therefore, our results clearly indicated that, although Br-Z1 has moderate ability to precociously induce *cut* transcription, Br-Z2 is the most likely isoform that governs Cut induction at the onset of metamorphosis.

We next sought to investigate how Br-Z2 synergized with TCF/Arm to induce precocious *cut* expression. We first carried out coimmunoprecipitation assay and found that both TCF and Arm were specifically coimmunoprecipitated with Br-Z2 from 293 T cell extracts (*Figure 6F,G*). To confirm the physical interaction between Arm and Br-Z2 in the nucleus, we next performed proximity ligation assay (PLA), which detects protein-protein interaction in situ with high specificity (*Söderberg et al., 2006*) (*Figure 6H*). Strong PLA signal was detected in the nuclei of S2 cells coexpressing Br-Z2 and Myc-tagged Arm (Arm-Myc) (*Figure 6I,J*). By contrast, PLA signal was barely detectable in S2 cells expressing Br-Z2 or Arm-Myc alone or coexpressing Br-Z2 and Myc-tagged ELL, a subunit of the transcription regulatory complex SEC (Super Elongation Complex) (*Figure 6I,J*) (*Liu et al., 2017*). These results clearly demonstrated that Arm and Br-Z2 physically interact within the nucleus. Furthermore, our detailed domain-mapping analysis revealed that the C-terminal domain but neither the ZF nor the BTB domain was crucial for Br-Z2 to physically interact with TCF (*Figure 6F,K*). Together, our results strongly suggested that Br-Z2 forms a transcription activation complex with TCF/Arm in inducing *cut* expression in future RPs.

## Identification of a distal intronic enhancer for the temporospatial induction of *cut*

We next sought to identify cis-regulatory elements of *cut* that confer its response to Wg and ecdysone signaling in future RPs. Since a *cut*-lacZ reporter harboring a well-characterized enhancer upstream of *cut* promoter (*Jack et al., 1991*; *Jia et al., 2016*) did not exhibit expression in RPs during metamorphosis, we systematically screened a series of *cut* enhancers. Out of 22 *cut* enhancer-Gal4 driver lines from the Janelia Gal4 collection, in which Gal4 is expressed under the control of *cut* enhancer fragments (*Pfeiffer et al., 2008*) (*Figure 7—figure supplement 1A*), we identified one line, R35B08-Gal4, which drove UAS-CD8-RFP expression specifically in future RPs at metamorphosis (*Figure 7—figure supplement 1B*). R35B08 is a previously-uncharacterized 3.2 kilobases (kb) enhancer fragment in the second intron of *cut*, approximately 50 kb downstream of *cut* promoter (cut-intron2-enhancer in *Figure 7A*). The temporal and spatial expression pattern of cut-intron2-

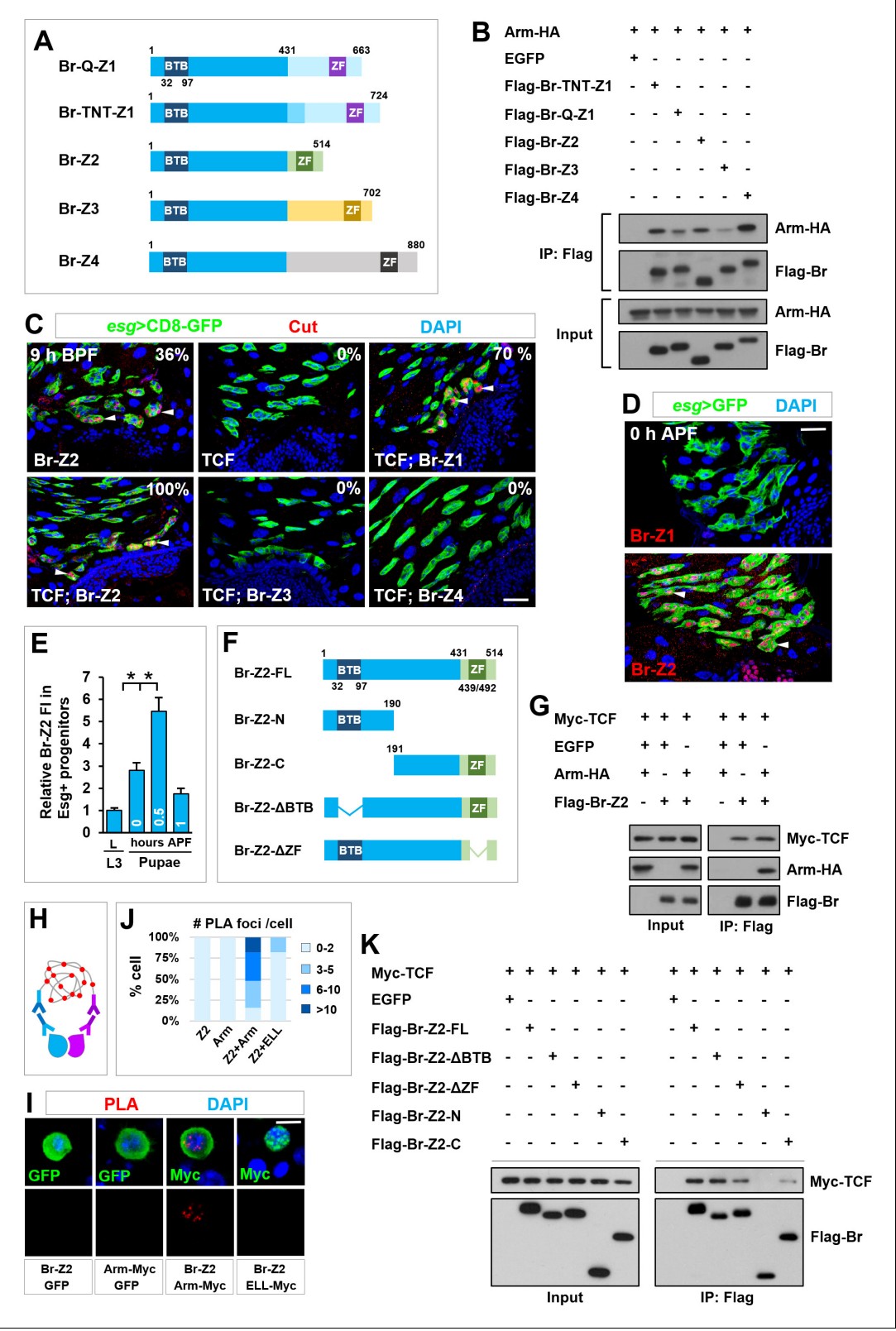

**Figure 6.** TCF/Arm physically associates with Br-Z2 in inducing *cut* transcription. (A) Schematic drawings of Br isoforms. Note that BTB and ZF indicate BTB (Broad-Complex, Tramtrack and Bric a brac) domain and Zinc Finger domain respectively. (B) Coimmunoprecipitation (CoIP) of Br isoforms and Arm in HEK293T cell extracts. In these and subsequent panels, GFP served as a negative control. (C) Cut expression pattern in Esg⁺ progenitors of indicated genotypes at 9–10 hr before puparium formation (BPF). (D) Expression pattern of Br-Z1 and Br-Z2 at midgut-hindgut boundary region at 0 hr

*Figure 6 continued on next page*

*Figure 6 continued*

APF. (E) Quantification of relative Br-Z2 fluorescent intensity (FI) in Esg$^+$ progenitors at different developmental stages. *p<0.0001 (n = 11–16). (F) Schematic drawings of Br-Z2 domains and truncated constructs. (G) CoIP of FLAG-tagged Br-Z2, HA-tagged Arm and Myc-tagged TCF in HEK293T cell extracts. (H–J) Specific Arm and Br-Z2 interaction detected by in situ PLA (proximity ligation assay) in the nuclei of *Drosophila* S2 cells. (H) Schematic diagram showing the principles of PLA. (I) S2 cells transfected with the indicated plasmids were detected for PLA signal (red). Note that the bottom panel shows single-channel images for PLA signal. Also note that ELL is a subunit of the Super Elongation Complex (SEC) that regulates gene transcription. Myc-tagged ELL serves here as a negative control. Quantification of the percentage of PLA foci number per transfected cell is shown in (J; n = 20–44). (K) CoIP of full-length (FL) or truncated FLAG-Br-Z2 and Myc-TCF. Scale bars, 25 µm (C, D) and 5 µm (I).
DOI: https://doi.org/10.7554/eLife.33934.017

The following source data and figure supplement are available for figure 6:

**Source data 1.** Input data for bar graph *Figure 6E,J*.
DOI: https://doi.org/10.7554/eLife.33934.019
**Figure supplement 1.** Dynamic expression of Br-Z2 and Br-Z1 during metamorphosis.
DOI: https://doi.org/10.7554/eLife.33934.018

enhancer-GFP, a GFP reporter that we generated for this intronic enhancer, was essentially identical to that of endogenous Cut protein (*Figure 7B*), indicating that the temporospatial induction of *cut* in future RPs is regulated at transcriptional level and primarily, if not solely, through this distal enhancer. Consistent with this idea, cut-intron2-enhancer-GFP expression in future RPs was highly responsive to alterations in Wg or ecdysone signaling (*Figure 7C,D*).

## TCF/Arm/Br-Z2 complex induces *cut* transcription likely via enhancer-promoter looping

The long distance between the intron2-enhancer and the promoter of *cut* suggested that chromatin looping might juxtapose the distal enhancer with *cut* promoter, crucial for *cut* induction. We therefore considered the tantalizing scenario whereby Br-Z2 self-association promotes the enhancer-promoter communication and *cut* transcription, based on the following observations: (1) *cut* intron2-enhancer contains closely-spaced putative TCF- and Br-Z2-binding sites (*Figure 7E*) (*Archbold et al., 2014*; *Chang et al., 2008*; *von Kalm et al., 1994*); (2) *cut* promoter region harbors putative Br-Z2-binding site but not TCF-binding site (*Figure 7E*); and (3) Br-Z2 contains BTB domain at its N-terminus (*Figure 6D*), which is likely to mediate protein dimerization or oligomerization (*Perez-Torrado et al., 2006*). To test this looping hypothesis, we first investigated whether TCF and Br-Z2 binds to their putative bindings site in the *cut* locus. Indeed, our electromobility shift assay (EMSA) results demonstrated a direct and sequence-specific binding of TCF and Br-Z2 to their putative binding sites in the *cut* promoter or intron2-enhancer region (*Figure 7F*). Next, we assessed whether Br-Z2 can self-associate. Our coIP data clearly showed that Br-Z2 formed protein dimer in a BTB domain-dependent manner (*Figure 7G*). Furthermore, deletion of only five amino acids (5AA; aa 46–50) in the BTB domain was sufficient to completely abolish the ability of Br-Z2 to self-associate (*Figure 7H,I*). Finally, we assayed the functional significance of Br-Z2 protein dimerization. Deletion of the whole BTB domain or only five amino acids in this domain (aa 46–50) abolished the activity of Br-Z2 to precociously induce *cut* expression within future RPs (*Figure 7J*), indicating that the ability to form protein dimer per se is crucial for Br-Z2 to control *cut* transcription. ZF domain-deleted form of Br-Z2 also failed to prematurely induce *cut* expression (*Figure 7J*), demonstrating that the sequence-specific DNA-binding ability is equally important for Br-Z2 to dictate *cut* transcription.

Taken together, our results identified the homeodomain protein Cut as a master switch that converts MPs into RPs in the right place at the right time (*Figure 8A*). RPs in turn migrate onto renal tubules and differentiate into renal cells (*Figure 8A*). When Cut is depleted, Esg$^+$ progenitors migrate normally, yet fail to switch identity and differentiate into midgut cells along renal tubules (*Figure 8A*). Mechanistically, the temporal and spatial signals inducing *cut* transcription in future RPs seem to intersect by facilitating enhancer-promoter looping of *cut*: At the onset of metamorphosis, the pulse of hormone ecdysone induced peak expression of Br. Br in turn acts as a transcription activator through its physical interaction with TCF/Arm and meanwhile likely serves as a looping factor juxtaposing the TCF/Arm-bound enhancer with *cut* promoter, triggering timely *cut* transcription. (*Figure 8B*).

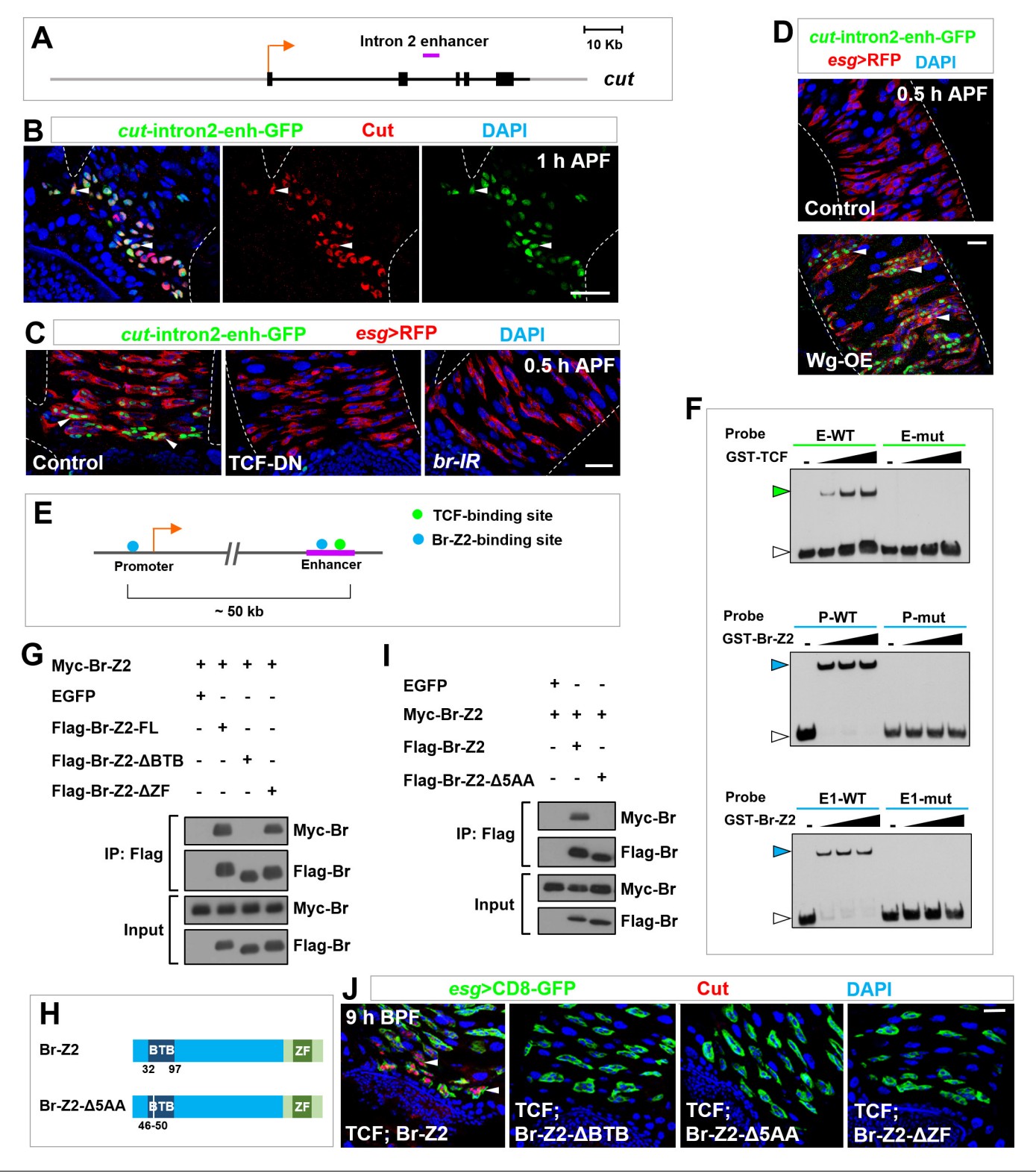

**Figure 7.** Self-association of Broad likely mediates enhancer-promoter looping of *cut* locus. (**A**) Schematic representation of the *cut* genomic locus. Exons, introns/non-coding regions and flanking genomic regions are represented as black bars, black lines and gray lines respectively. Purple bar indicates intron2-enhancer. (**B**) cut-intron2-enhancer-GFP, a reporter for the intron2 enhancer of *cut*, specifically colocalizes with endogenous Cut in future RPs. (**C**) The expression of cut-intron2-enhancer-GFP is abolished upon expression of TCF-DN or *br-RNAi* in Esg⁺ progenitors. (**D**) cut-intron2-enhancer-GFP is ectopically induced in MPs far away from the midgut-hindgut boundary upon Wg overexpression. (**E**) Schematic drawing of a portion

*Figure 7 continued on next page*

*Figure 7 continued*

of the *cut* locus spanning approximately 60 kb. Putative TCF-binding sites and Br-Z2-binding sites are represented by green and blue dots respectively. Purple line indicates intron2-enhancer. (**F**) EMSA demonstrating the interaction of TCF or Br-Z2 DNA-binding domains with biotin-labelled probes bearing wild type (WT) or mutant (mut) sequence of putative TCF- or Br-Z2-binding sites. E: enhancer; P: promoter. Note that green or blue arrowheads indicate the DNA-protein complexes, whereas white arrowheads indicate free probes. (**G**) CoIP between full-length (FL) or truncated FLAG-tagged Br-Z2 and Myc-tagged Br-Z2 in 293T cells. Note that dimerization of Br-Z2 depends on its BTB but not ZF domain. (**H**) Schematic drawings of Br-Z2-Δ5AA construct, a small deletion in the BTB domain. (**I**) CoIP between Flag-Br-Z2-Δ5AA and Myc-Br-Z2 in 293T cells. (**J**) Coexpression of TCF with Br-Z2-ΔBTB, Br-Z2-Δ5AA or Br-Z2-ΔZF failed to precociously induce Cut expression at 9 hr BPF. Scale bars, 25 μm.
DOI: https://doi.org/10.7554/eLife.33934.020

The following figure supplements are available for figure 7:

**Figure supplement 1.** Identification of an intronic enhancer conferring temporospatial induction of *cut* in future RPs.
DOI: https://doi.org/10.7554/eLife.33934.021

**Figure supplement 2.** EMSA assay showing specific binding of TCF or Br-Z2 to their binding site in *cut* promoter or intron2-enhancer region.
DOI: https://doi.org/10.7554/eLife.33934.022

One important predication of this model is that cohesin (*Dorsett, 2011*; *Dorsett and Merkenschlager, 2013*; *Dowen and Young, 2014*) and its loading factor Nipped-B, which facilitates and stabilizes enhancer-promoter looping (*Rollins et al., 1999*), are essential for *cut* induction in future RPs. Indeed, *cut* expression at metamorphosis was markedly decreased upon depletion of cohesin subunit Stromalin (SA) or Nipped-B (*Figure 8C*). Significantly, in sharp contrast, the expression of Fz3-GFP (*Figure 8D*) or cut-intron2-enhancer-GFP (*Figure 8E*) remained unaltered upon downregulation of cohesin or Nipped-B, indicating that cohesin is specifically required for chromatin looping-dependent *cut* transcription in future RPs.

## Discussion

### Homeobox protein Cut dictates a unique and natural lineage conversion event

Here we revealed a naturally-occurring midgut-to-renal lineage conversion event at the onset of *Drosophila* metamorphosis. Compared with experimentally induced reprogramming, natural reprogramming events in physiological settings are relatively rare, yet much more efficient, predictable and robust (*Gettings et al., 2010*; *Jarriault et al., 2008*; *Red-Horse et al., 2010*; *Schaub et al., 2015*). The lineage reprogramming process unveiled in our studies represents a unique natural lineage conversion event in that (1) The cell identity switch occurs between organ-specific progenitors, not fully-differentiated cells; (2) the reprogramming event takes place at postembryonic stages, when cells are much less plastic; and (3) the cell identity fully converts from one organ-specific characteristics to another. Thus, this midgut-to-renal lineage conversion event provides a previously unexplored physiological context for elucidating the detailed molecular mechanisms underlying cell plasticity.

Our results further show that the homeodomain protein Cut is a master cell identity switch controlling the natural conversion of midgut progenitors into renal identity. Cut was originally identified as a binary identity switch between subtypes of neurons in the peripheral nervous systems (PNS) of *Drosophila* (*Bodmer et al., 1987*). Compelling evidence demonstrated that Cut is both necessary and sufficient in specifying neuronal identities in fly PNS (*Blochlinger et al., 1991*; *Bodmer et al., 1987*). Therefore, Cut dictates cell identity switch within or across organ boundary.

We reason that Cut might be particularly suitable for being a master cell identity switch in diverse biological contexts. Firstly, *cut* is an unusually large gene harboring a long and complex enhancer region spanning more than 150 kb. Such a long enhancer region can be subdivided into small segments responsive to different stimuli or signals in diverse tissues or organs at distinct developmental stages. Therefore, akin to neurons with extensive and complex arbors, the extra-long and segmented enhancer region of *cut* receives and integrates diverse input signals, and drives Cut expression with high temporospatial precision. Secondly, Cut, as a homeobox transcription factor, may intrinsically possess the ability to specify and confer organ or tissue identities, analogous to the classic homeotic genes such as *Antennapedia*, by simultaneously erasing old cell identities and writing new ones.

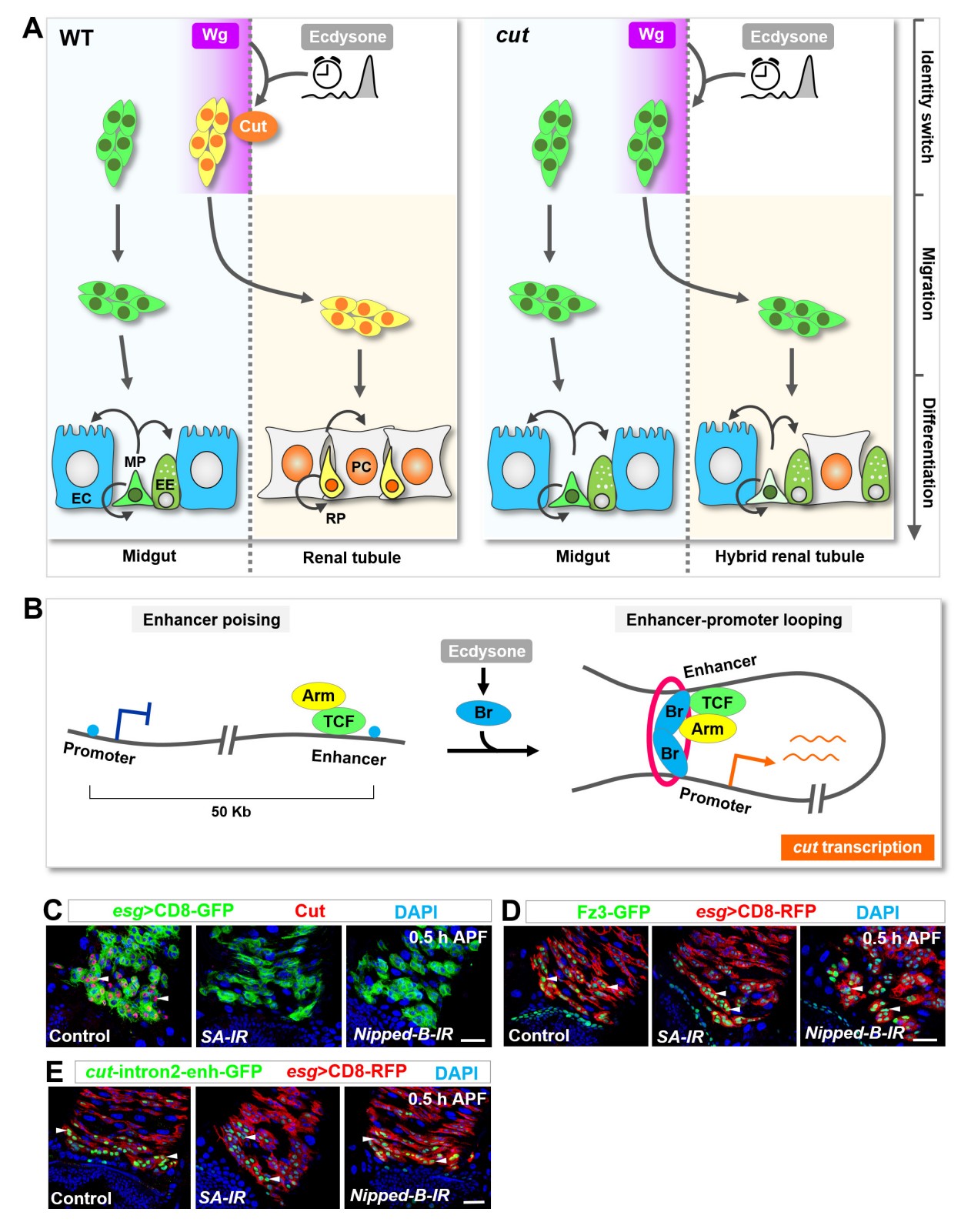

**Figure 8.** Ecdysone signaling and Wg pathway might intersect through a poising-and-bridging mechanism to dictate midgut-to-renal lineage reprogramming. (**A**) A graphic model of the MP-RP identity switch. Cut (orange), induced by Wg morphogen gradient (purple; spatial cue) in conjunction with the steroid hormone ecdysone (grey; temporal cue), dictates midgut-to-renal progenitor identity switch at the onset of metamorphosis. (**B**) A working model depicting the poising-and-bridging mechanism. Note that the red circle represents cohesin, which is likely to be

*Figure 8 continued*

important for stabilizing the enhancer-promoter looping of *cut*. (**C–E**) Downregulation of either cohesin subunit SA or cohesin loading factor Nipped-B abolished *cut* induction (**C**) but showed no effects on Fz3-GFP (**D**) or cut-intron2-enhancer-GFP (**E**) expression in future RPs. Scale bars, 25 µm.

DOI: https://doi.org/10.7554/eLife.33934.023

## Precise induction of master identity switch by a 'poising-and-bridging' mechanism

Acquisition and switch of distinct cell identities are precisely and tightly controlled in both space and time (*Erclik et al., 2017*; *Red-Horse et al., 2010*; *Schaub et al., 2015*). Yet the molecular basis underlying the integration of spatial and temporal signals remain poorly understood. We posit that two molecular mechanisms may underlie the intersection of temporal and spatial cues: (1) the spatial cues restrict the expression/activity of the temporal pathway component(s), or vice versa. This strategy has been well exemplified in spatiotemporal control of border cell migration in the *Drosophila* ovary (*Jang et al., 2009*) and spatial restriction of neural competence (*Huang et al., 2014*); and (2) the temporal and spatial pathways converge to induce the expression of a specific set of target genes. The molecular basis of the latter scenario remains enigmatic.

Interestingly, our results revealed that the spatial and temporal cues might intersect by an unexpected 'poising and bridging' mechanism to dictate the midgut-to-renal progenitor identity switch (*Figure 8B*). In this model, the spatial transcription factors (TF) bind a distal enhancer and prime it for a timely response to the temporal cues, whereas the temporal TF acts as a bridging factor that binds to both the distal enhancer and the promoter region and induces enhancer-promoter looping through its self-association (*Figure 8B*). Meanwhile, the temporal TF is likely to also act as a transcription activator by physically interacting with the spatial TFs. Importantly, since protein dimerization or oligomerization occurs only when the protein concentration rises above certain threshold (*Marianayagam et al., 2004*), such a protein-dimerization-based regulatory mechanism is ideal for integrating and translating gradual changes in temporal and spatial signaling strength into a timely and all-or-none biological event such as cell identity switch.

Although long-range chromatin looping has been found at numerous gene loci (*Ghavi-Helm et al., 2014*; *Levine et al., 2014*), the identity and mode of action of looping factors under developmental or physiological settings remain unclear. Our findings shed mechanistic insights into how a temporal factor might bridge the distal enhancer and the promoter of a master gene via protein dimerization in development. In light of recent studies implicating dimerization of TFs, such as CTCF, Yin Yang one and LDB1, in the organization of chromatin architecture either globally (*Phillips and Corces, 2009*; *Weintraub et al., 2017*) or locally (*Deng et al., 2012*), it is conceivable that TF dimerization or multimerization might represent a precise and prevailing mechanism establishing chromatin loops in space and time.

# Materials and methods

**Key resources table**

| Reagent type (species) or resource | Designation | Source or reference | Identifiers | Add. info. |
|---|---|---|---|---|
| Genetic reagent (*D. melanogaster*) | UAS-cut | Bloomington Drosophila Stock Center | RRID: BDSC_36496 | |
| Genetic reagent (*D. melanogaster*) | FRT19A, cut[C145] | Bloomington Drosophila Stock Center | RRID: BDSC_36496 | |
| Genetic reagent (*D. melanogaster*) | FRT19A, dsh[3] | Bloomington Drosophila Stock Center | RRID: BDSC_6331 | |
| Genetic reagent (*D. melanogaster*) | UAS-Wg-HA | Bloomington Drosophila Stock Center | RRID: BDSC_5918 | |
| Genetic reagent (*D. melanogaster*) | UAS-Arm-S10 | Bloomington Drosophila Stock Center | RRID: BDSC_4782 | |

*Continued on next page*

*Continued*

| Reagent type (species) or resource | Designation | Source or reference | Identifiers | Add. info. |
|---|---|---|---|---|
| Genetic reagent (*D. melanogaster*) | esg-Gal4, tubP-Gal80<sup>ts</sup>, UAS-GFP | Gift from Dr. H Jasper (*Biteau et al., 2008*) and Dr. C Micchelli (*Micchelli and Perrimon, 2006*) | N/A | |
| Genetic reagent (*D. melanogaster*) | wg-Gal4 | Gift from Dr. JP Vincent (*Alexandre et al., 2014*) | N/A | |
| Genetic reagent (*D. melanogaster*) | wg(KO; NRT–Wg) | Gift from Dr. JP Vincent (*Alexandre et al., 2014*) | N/A | |
| Genetic reagent (*D. melanogaster*) | UAS-TCF-DN | Bloomington Drosophila Stock Center | RRID: BDSC_4785 | |
| Genetic reagent (*D. melanogaster*) | wg-lacZ | Bloomington Drosophila Stock Center | RRID: BDSC_1672 | |
| Genetic reagent (*D. melanogaster*) | Rab3-GFP | Bloomington Drosophila Stock Center | RRID: BDSC_62541 | |
| Genetic reagent (*D. melanogaster*) | UAS-CD8-RFP | Bloomington Drosophila Stock Center | RRID: BDSC_27392 | |
| Genetic reagent (*D. melanogaster*) | cut-intron2-Gal4 | Bloomington Drosophila Stock Center | RRID: BDSC_49818 | |
| Genetic reagent (*D. melanogaster*) | NRE-GFP | Bloomington Drosophila Stock Center | RRID: BDSC_30727 | |
| Genetic reagent (*D. melanogaster*) | UAS-Sgg.S9A | Bloomington Drosophila Stock Center | RRID: BDSC_5255 | |
| Genetic reagent (*D. melanogaster*) | UAS-EcR-DN | Bloomington Drosophila Stock Center | RRID: BDSC_6872 | |
| Genetic reagent (*D. melanogaster*) | UAS-Br-Z1 | Bloomington Drosophila Stock Center | RRID: BDSC_51190 | |
| Genetic reagent (*D. melanogaster*) | UAS-Br-Z3 | Bloomington Drosophila Stock Center | RRID: BDSC_51192 | |
| Genetic reagent (*D. melanogaster*) | UAS-Br-Z4 | Bloomington Drosophila Stock Center | RRID: BDSC_51193 | |
| Genetic reagent (*D. melanogaster*) | UAS-TCF | Bloomington Drosophila Stock Center | RRID: BDSC_4838 | |
| Genetic reagent (*D. melanogaster*) | UAS-cut-IR | Bloomington Drosophila Stock Center | RRID: BDSC_33967 | |
| Genetic reagent (*D. melanogaster*) | UAS-br-IR | Bloomington Drosophila Stock Center | RRID: BDSC_27272 | |
| Genetic reagent (*D. melanogaster*) | UAS-Nipped-b-IR | Bloomington Drosophila Stock Center | RRID: BDSC_32406 | |
| Genetic reagent (*D. melanogaster*) | UAS-SA-IR | Bloomington Drosophila Stock Center | RRID: BDSC_33395 | |
| Genetic reagent (*D. melanogaster*) | UAS-wg-IR | Bloomington Drosophila Stock Center | RRID: BDSC_33902 | |
| Genetic reagent (*D. melanogaster*) | UAS-Eip74EF-IR | Bloomington Drosophila Stock Center | RRID: BDSC_29353 | |
| Genetic reagent (*D. melanogaster*) | UAS-Eip75B -IR | Bloomington Drosophila Stock Center | RRID: BDSC_26717 | |
| Genetic reagent (*D. melanogaster*) | UAS-white-IR | Bloomington Drosophila Stock Center | RRID: BDSC_33623 | |
| Antibody | Anti-Flag M2 affinity gels | Sigma-Aldrich | Cat#: A2220 | |
| Antibody | Mouse anti-Broad-core (25E9.D7) | Developmental Studies Hybridoma Bank | RRID: AB_528104 | |
| Antibody | Mouse anti-Broad-Z1 (Z1.3C11.OA1) | Developmental Studies Hybridoma Bank | RRID: AB_528105 | |

*Continued on next page*

*Continued*

| Reagent type (species) or resource | Designation | Source or reference | Identifiers | Add. info. |
|---|---|---|---|---|
| Antibody | Mouse anti-Cut (2B10) | Developmental Studies Hybridoma Bank | RRID: AB_528186 | |
| Antibody | Mouse anti-Prospero (MR1A) | Developmental Studies Hybridoma Bank | RRID: AB_528440 | |
| Antibody | Mouse anti-Bruchpilot (nc82) | Developmental Studies Hybridoma Bank | RRID: AB_2314866 | |
| Antibody | Mouse anti-Allatostatin (Ast7, 5F10) | Developmental Studies Hybridoma Bank | RRID: AB_528076 | |
| Antibody | Rabbit anti-Pdm1 | Gift from Dr. X. Yang | N/A | |
| Antibody | Anti-GFP antibody | abcam | Cat#: ab13970 | |
| Antibody | Anti-GFP antibody - ChIP Grade | abcam | Cat#: ab290 | |
| Antibody | Mouse anti-beta-galactosidase (40-1a) | Developmental Studies Hybridoma Bank | RRID: AB_2314509 | |
| Antibody | Anti-RFP antibody | abcam | Cat#: ab62341 | |
| Antibody | Phospho-Histone H3 (Ser10) Antibody | Cell Signaling Technology | Cat#: 9701 | |
| Antibody | Rabbit anti-Myc | Cell Signaling Technology | Cat#: 2278 | |
| Commercial assay or kit | Duolink In Situ Red Starter Kit Mouse/Rabbit | Sigma-Aldrich | Cat#: DUO92101 | |
| Strain, strain background (*E.coli*) | BL21 (DE3) | TransGen Biotech | Cat#: CD601-02 | |
| Software, algorithm | Photoshop CS5 | Adobe | N/A | |
| Software, algorithm | The Leica Application Suite 2.6.3 | Leica | N/A | |
| Cell line (Human) | HEK293T | ATCC | RRID: CRL-3216 | |
| Cell line (*D. melanogaster*) | S2 | DGRC | Cat#: S2-DGRC | |
| Recombinant DNA reagent | pcDNA3.1 | Invitrogen | Cat#: V79020 | |
| Recombinant DNA reagent | pAc5.1 | Invitrogen | Cat#: V4110-20 | |
| Recombinant DNA reagent | pGEX-6P-1 | GE Healthcare | Cat#: 28-9546-48 | |

## Fly genetics

Fly culcture and crosses were performed according to standard procedures. *Drosophila* stocks used in this study include: UAS-Cut (#36496; Bloomington stock center (BDSC)); FRT19A, *cut*[C145] (#36496; BDSC); UAS-Wg-HA (#5918; BDSC); UAS-Arm-S10 (#4782; BDSC); UAS-cut-RNAi (#33967; BDSC); UAS-br-RNAi (#27272; BDSC); *esg*-Gal4, tubP-Gal80[ts], UAS-GFP (*Biteau et al., 2008*; *Micchelli and Perrimon, 2006*); *wg*-Gal4 (*Alexandre et al., 2014*); *wg(KO; NRT–Wg)* (*Alexandre et al., 2014*); UAS-TCF-ΔN (#4785; BDSC); *wg*-lacZ (#1672; BDSC); Rab3-GFP (#62541; BDSC); UAS-CD8-RFP (#27392, BDSC); *cut*-intron2-Gal4 (#49818, BDSC); NRE-GFP (#30727; BDSC); UAS-Sgg.S9A (#5255; BDSC); UAS-EcR-DN (#6872; BDSC); UAS-wg-RNAi (#33902; BDSC); UAS-Br-Z1 (#51190; BDSC); UAS-Br-Z3 (#51192; BDSC); UAS-Br-Z4 (#51193; BDSC); FRT19A, *dsh*[3] (#6331; BDSC); UAS-TCF (#4838; BDSC); UAS-Nipped-B-RNAi (#32406; BDSC); UAS-SA-RNAi (#33395; BDSC); UAS-Eip74EF-RNAi (#29353; BDSC); UAS-Eip75B (#26717; BDSC); UAS-white-RNAi (#33623; BDSC); Act5C-FRT>stop>FRT-lacZ.nls (#6355; BDSC).

*wg*-Gal4 transgenic flies were generated by integrating the Gal4 containing plasmid RIVGal4 (*Baena-Lopez et al., 2013*) into the attP site of wg flies (*Alexandre et al., 2014*).

## Molecular biology

To generate Fz3-pH-Stinger and cut-intron2-enhancer-pH-Stinger constructs, a 2,318 bp genomic DNA fragment (−2324 to −7) from *frizzled*3 gene region and a 3,157 bp genomic DNA fragment from *cut* gene region were PCR amplified and inserted into the pH-Stinger vector respectively. Transgenic lines of these reporters were generated by site-specific integration into the attP2 landing site on the third chromosome using standard phiC31 transformation methods. All transgenic plasmids were verified by DNA sequencing before germline transformation.

The PCR primers are listed as below: *fz* three reporter Fw: 5′-GAACGAAAGAGTTGGCAGAGAG −3′; *fz* three reporter Rv: 5′-GCTTAGTGGGTTTCAGGAGG−3′; *ct* reporter Fw: 5′-GCCGAGA TGCGGTAGTAAAACG−3′; *ct* reporter Rv: 5′-CTGTTTGTTTCTGGCGAGCTTA −3′.

To generate UAS-FLAG-Br-Z2-FL; UAS-FLAG-Br-Z2-ΔBTB; UAS-FLAG-Br-Z2-Δ5AA or UAS-FLAG-Br-Z2-ΔZF transgenic lines, full length or truncated version of FLAG-Br-Z2 was inserted into pUAST vector, followed by phiC31-mediated integration into the attP2 landing site.

For Coimmunoprecipitation experiments, Arm-HA in pcDNA3.1 (Invitrogen) was made by cloning Arm-S10 from genome DNA extracted from UAS-Arm-S10 transgenic fly line (#4782; BDSC), followed by replacing Myc tag with HA tag (YPYDVPDYA). FLAG-tagged Br isoforms in pcDNA3.1 were constructed by cloning cDNA of each isoform from corresponding br isoform-specific transgenic lines using genomic DNA PCR. aa 191–514; aa 1–190; aa 32–97, aa 439–492 and aa 46–50 of Br-Z2 were deleted to make Br-Z2-N, Br-Z2-C, Br-Z2-ΔBTB, Br-Z2-ΔZF and Br-Z2-Δ5AA respectively. pCMV-Myc-TCF is a generous gift from Dr. Esther M Verheyen.

## MARCM clonal analysis

To generate MARCM clones shown in *Figure 1H–I′*, *Figure 3I,J* or *Figure 4—figure supplement 1E*, larvae were heat-shocked at 37°C for 6 times for 40 min each time successively at 24, 28, 48, 52, 72, 76 hr after-larvae-hatching (ALH) and further aged at 25°C before dissection at early adult stage (2–3 days after eclosion).

To generate MARCM clones shown in *Figure 3G,H*, larvae were heat-shocked at 37°C for 6 times for 60 min each time successively at 24, 28, 48, 52, 72, 76 hr after-larvae-hatching (ALH) and further aged at 25°C before dissection at 72 hr APF.

To generate MARCM clones shown in *Figure 3M*, third instar larvae were heat-shocked at 37°C for 40 min and farther aged at 25°C before dissection at 96 hr APF.

## Lineage-tracing

The lineage tracing experiments as shown in *Figure 3A–F′* were performed by crossing the *esg*-Gal4, Gal80$^{ts}$, UAS-CD8-GFP; UAS-w-RNAi or *esg*-Gal4, Gal80$^{ts}$, UAS-CD8-GFP; UAS-*cut*-RNAi flies with UAS-FLP; Act5C-FRT>stop>FRT-lacZ.nls flies. Embryos were collected and kept at 18°C. Late third instar larvae were shifted to 29°C until dissection at early adult stage (2–3 days after eclosion).

## Immunohistochemistry

For intestine-renal tubule immunostaining, samples were dissected in Schneider's insect medium (Sigma-Aldrich) and proceeded as previously described (*Lin et al., 2008*) with modifications. Briefly, samples were fixed in 4% formaldehyde/1xPBS/n-heptane (v/v/v = 1:1:2) for 15 min at room temperature, washed with 100% methanol and gradually rehydrated in 75, 50 and 25% PBST (1xPBS plus 0.1% Triton X-100)/methanol mix. Samples were washed several times with PBST, blocked in blocking solution (1% BSA in PBST) for 1 hr, followed by incubating with appropriate primary antibody overnight at 4°C. After incubation with secondary antibodies according to standard procedures, samples were mounted in Vectashield (Vector Laboratories). For DNA staining, Hoechst (Life Technologies) was added in the wash step with a dilution of 1:3000. Images were obtained on a Leica TCS SP8 AOBS confocal microscope and were processed with Adobe Photoshop.

Primary antibodies used in this study were chicken anti-GFP (1:2000, Abcam), mouse anti-Pros (1:100, Developmental Studies Hybridoma Bank [DSHB]), mouse anti-Cut (2B10) (1:100, DSHB), mouse anti-Broad-core (25E9) (1:200, DSHB), mouse anti-Nc82 (1:100, DSHB), mouse anti-AstA (1:100, DSHB), rabbit anti-Pdm1 (1:1000, a generous gift from Dr. Xiaohang Yang), mouse anti-β-galactosidase (1:100, DSHB), rabbit anti-β-galactosidase (1:1000, Cappel), rabbit anti-phospho-Histone H3 (1:1000, Upstate).

## Cell lines and transfection

Human embryonic kidney HEK293T cells (ATCC, RRID: CRL-3216 obtained from Dr. Hong Wu's laboratory, Peking University, and authenticated by ATCC) were maintained in DMEM medium (Invitrogen) supplemented with 10% FBS at 37 °C and 5% CO2. DNA transfection was performed using a standard polyethylenimine (PEI) protocol.

Drosophila S2 (Schneider 2) cells (DGRC, Cat#: S2-DGRC; obtained from Dr. Alan Jian Zhu's laboratory, Peking University, and authenticated by DGRC) were cultured at 25°C in Schneider's Drosophila medium Drosophila Medium (Invitrogen) supplemented with 10% FBS, 100 U/ml penicillin and 100 mg/ml streptomycin. DNA transfection of S2 cells were carried out using Effectene Transfection Reagent (QIAGEN) according to manufacturer's protocol.

Both cell lines used in this study have been tested for and confirmed to be negative for mycoplasma contamination, using short tandem repeat (STR) profiling technique.

## Coimmunoprecipitation

Coimmunoprecipitation assays in HEK293T cell extracts were performed as previously described (*Liu et al., 2017*). Briefly, 48 hr after transfection, HEK293T cells were harvested, washed and resuspended in lysis buffer [50 mM Tris-HCl (pH 8.0); 120 mM NaCl; 5 mM EDTA; 1% NP-40; 10% glycerol; protease inhibitor cocktail (Sigma-Aldrich); 2 mM $Na_3VO_4$] and kept on ice for 20 min. Cell extracts were sonicated with Bioruptor Plus (Biosense) at 4°C with low power for 5 cycles of 10 s on/10 s off. The cell extracts were clarified by centrifugation, and proteins immobilized by binding to anti-FLAG M2 (Sigma-Aldrich) affinity gel for 4 hr at 4°C. Beads were washed and proteins recovered directly in SDS-PAGE sample buffer. Rabbit anti-Flag (Cell Signaling Technology), Rabbit anti-c-myc (Cell Signaling Technology) or rabbit anti-HA (Santa Cruz Biotechnologies) were used for western blot analysis.

## Br-Z2 antibody

Isoform-specific rabbit anti-Br-Z2 antibody was generated in this study [GST fusion of Br-Z2 aa 432–514, affinity purified (Abclonal Biotech.)] and used at 1:40 for immunostaining.

## Protein purification

The DNA-binding domains (DBD) of TCF (aa 271–408) and Br-Z2 (aa 432–514) were cloned into pGEX-6P-1. GST-tagged protein was purified by ProteinIso GST resin (Transgen Biotech) through column buffer (25 mM HEPES (pH 7.6), 150 mM NaCl, 10% glycerol, 1 mM EDTA, 10 mM β-mercaptoethanol, 2 mM PMSF). After washing, protein was eluted with 100 mM glutathione in column buffer. Protein concentrations were measured by Coomassie stained gels. Known concentrations of BSA (bovine serum albumin) were used as a standard.

## Electromobility shift assays

Increasing concentration of purified GST-tagged protein and 10 fmol biotin-labelled double-stranded DNA substrate were incubated at 25°C in 20 µl reaction buffer (20 mM HEPES (pH 7.9), 50 mM KCl, 0.1 mM EDTA, 2 mM DTT, 6 mM $MgCl_2$, 0.1 mg/ml BSA, 50 ug/ml poly(dI-dC), 5% glycerol) for 45 min. The reaction mixture was loaded and resolved in 8% TBE gel. Amounts of recombinant protein used per reaction were as follows: 1.2–7.2 ug GST-TCF-DBD (E probe); 0.01–0.2 ug GST-Br-Z2-DBD (P probe) and 0.2–1 ug GST-Br-Z2-DBD (E1 probe).

## Proximity ligation assay (PLA)

Duolink in situ PLA was performed with *Drosophila* S2 cells according to manufacturer's instructions (DUO92101; Sigma-Aldrich). Briefly, after transfection and fixation, S2 cells were incubated with primary antibodies at RT for 3 hr, followed by incubation with Duolink PLA probes (1:12) at 37°C for 1 hr, ligation at 37°C for 1 hr and amplification at 37°C for 2 hr. Primary antibodies used were rabbit anti-Myc (1:200) and mouse-anti-Br-core (1:150).

## Quantification and statistical analysis

Length of ureter was measured with Leica Application Suite 2.6.3 from Leica Microsystems. For quantification of the intensity of antibody staining, images were taken with the same confocal

settings and the mean fluorescence intensity was measured with Histogram function of Adobe Photoshop. Unpaired Student's t-tests were used for statistical analysis between two groups.

## Acknowledgements

We are grateful to Drs. Heinrich Jasper, Renjie Jiao, Liqun Luo, Craig Micchelli, Yi Rao, Esther M Verheyen, Jean-Paul Vincent, Hong Wu, Xiaohang Yang, Alan Jian Zhu, University of Iowa DSHB, VDRC, Bloomington *Drosophila* Stock Center, and the TRiP at Harvard Medical School (NIH/NIGMS R01-GM084947) and Tsinghua University for fly stocks and reagents; Dr. Kang Shen for helpful discussion; and members of the Song lab for help. This work was supported by grants from the National Natural Science Foundation of China [31471372 and 31771629], Ministry of Education Key Laboratory of Cell Proliferation and Differentiation and the Peking-Tsinghua Center for Life Sciences awarded to YS.

## Additional information

### Funding

| Funder | Grant reference number | Author |
|---|---|---|
| National Natural Science Foundation of China | 31471372 | Yan Song |
| Peking-Tsinghua Center for Life Sciences | | Yan Song |
| Ministry of Education Key Laboratory of Cell Proliferation and Differentiation | | Yan Song |
| National Natural Science Foundation of China | 31771629 | Yan Song |

The funders had no role in study design, data collection and interpretation, or the decision to submit the work for publication.

### Author contributions

Ke Xu, Conceptualization, Data curation, Formal analysis, Methodology, Writing—original draft, Writing—review and editing; Xiaodan Liu, Data curation, Formal analysis, Methodology; Yuchun Wang, Chouin Wong, Data curation, Formal analysis; Yan Song, Conceptualization, Formal analysis, Supervision, Funding acquisition, Writing—original draft, Writing—review and editing

### Author ORCIDs

Yan Song http://orcid.org/0000-0002-1413-6123

### Decision letter and Author response

Decision letter https://doi.org/10.7554/eLife.33934.026
Author response https://doi.org/10.7554/eLife.33934.027

## Additional files

### Supplementary files

• Transparent reporting form
DOI: https://doi.org/10.7554/eLife.33934.024

### Data availability

All data generated or analysed during this study are included in the manuscript and supporting files.

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
