## [Decision Letter]

Thank you for submitting your article "Temporospatial induction of homeodomain gene *cut* via chromatin looping dictates natural lineage reprogramming" for consideration by *eLife*. Your article has been reviewed by two peer reviewers, and the evaluation has been overseen by a Reviewing Editor and K VijayRaghavan as the Senior Editor. The reviewers have opted to remain anonymous.

The reviewers have discussed the reviews with one another and the Reviewing Editor has drafted this decision to help you prepare a revised submission.

Summary:

Xu and colleagues make an important contribution on a form of cell identity reprogramming that depends on the intersection of spatial information dictated by Wg signaling and a temporal cue downstream of ecdysone signaling during metamorphosis. Intersection of these cues leads to induction of the Cut homeodomain transcription factor and conversion of midgut progenitors to a renal progenitor identity. In addition to the insight into a mechanism of temporospatial control of cell identity specification, insights gained here regarding factors that control transcription factor expression may be broadly applicable to tissue reprogramming and stem cell biology.

Through immunofluorescence, the authors describe the molecular regulation of, and the role of, the cut gene during midgut to renal lineage conversion in *Drosophila*. Cut and related genes had already been implicated in kidney development. Spatiotemporal regulation of the cut gene is now described here, where during metamorphosis some Esg^+^ progenitor cells migrate from the midgut to the renal tubules where they express Cut to give rise to renal progenitor cells.

Immunofluorescent analyses found that Cut is not necessary for midgut to renal tubule cell migration. The loss of Cut was however lethal to the majority of flies. Escaper flies lacked the lower ureter region and gave rise to intestinal Pros+ Cut- EE cells in the renal tubules. These anomalies suggest that Cut modulates a key renal developmental step by acting as a master identity switch. Cut expression in APF cells near the midgut-hindgut boundary is minimally regulated by Wg and ecdysone signalling. Further experiments identify a novel enhancer element and a promoter-enhancer looping model is proposed.

The role of Cut throughout fly development is extensively shown. This part of the proposed model is supported by extensive images of developing *Drosophila* and makes use of robust genetic and cellular tools. These results alone may warrant publication.

Essential revisions:

Specific important comments that need to be addressed.

1) The data to support the conclusion in the paper are strong, but an exception is the data to support a Broad-dependent promoter-enhancer looping mechanism. Although perfectly plausible (and even likely), the looping model is not well supported by the experiments. In order to support this idea to the extent that the authors claim, we think additional experiments would be required, such as a more direct experimental measure of the ability of Broad to loop DNA to pin down this proposed mechanism. If these experiments are not speedily feasible the conclusion on this looping mechanism should be alluded to in the Discussion section but removed from the Title and not presented in the text as bona fide.

2) Other experiments (such as the RNAi screen, and the identification of a new enhancer element) are short on detail and this part should be addressed.

3) Another substantial difficulty was that the inferences regarding developmental transitions that occur at the onset of metamorphosis. These were often difficult to follow and not written in a way that is easily grasped by non-experts. The same is true for the G-TRACE results. The authors run through these data fairly rapidly without making interim conclusions or putting their observations into context. There is a lot for the reader to keep track of cell name, cell position in the different organs, location of different organs and organ parts, ploidy, cut expression status, Esg status, developmental stage, and potentially unfamiliar acronyms build up quickly. Since this description sets up the remainder of the paper, it is important to rewrite these sections with a general reader in mind to help increase the appeal to the readership of *eLife*.

---

## [Author Response]

Essential revisions:Specific important comments that need to be addressed.1) The data to support the conclusion in the paper are strong, but an exception is the data to support a Broad-dependent promoter-enhancer looping mechanism. Although perfectly plausible (and even likely), the looping model is not well supported by the experiments. In order to support this idea to the extent that the authors claim, we think additional experiments would be required, such as a more direct experimental measure of the ability of Broad to loop DNA to pin down this proposed mechanism. If these experiments are not speedily feasible the conclusion on this looping mechanism should be alluded to in the Discussion section but removed from the Title and not presented in the text as bona fide.

We wish to thank the editors and reviewers for all the efforts they have put into reviewing and improving our manuscript. We agree with the reviewers that more direct evidence supporting the Broad-dependent-looping model, such as a 3C (chromatin confirmation capture) experiment using FACS-purified future renal progenitors at P0 stage, would be ideal. This is indeed some experiment that we have been thinking about as well. However, due to the small number of future renal progenitors in each fly pupal midgut, it would be too time-consuming and technically challenging to complete such 3C experiment in a timely manner. Therefore, we decided to follow reviewers’ insightful suggestions and tone down the language on this looping mechanism. The manuscript has been modified as following: (1) remove “via chromatin looping” from the title; (2) move the working model of cut enhancer-promoter looping to the Discussion section; (3) tone down our statements on this looping model throughout the text. As an example, “Our results therefore led us to propose an unexpected poising-and-bridging mechanism whereby spatial and temporal cues intersect via chromatin looping to turn on a master transcription factor and dictate efficient and precise lineage reprogramming” has been changed to “Our results therefore led us to propose an unexpected poising-and-bridging mechanism whereby spatial and temporal cues intersect, likely via chromatin looping, to turn on a master transcription factor and dictate efficient and precise lineage reprogramming” (Abstract).

2) Other experiments (such as the RNAi screen, and the identification of a new enhancer element) are short on detail and this part should be addressed.

Thank you for this important point. For the RNAi screen, we have now added one paragraph describing the strategy and detailed procedure of this genome-wide RNAi-based screen (subsection “Adult renal stem cells specifically express homeodomain protein Cut”), as well as a supplemental figure panel (Figure 1—figure supplement 1B) demonstrating the cut-RNAi-induced renal tubule phenotype that we initially observed while carrying out the screen.

As for the identification of the new cut enhancer element, we have now added a few sentences describing how the new enhancer was found: “Out of 22 cut enhancer-Gal4 driver lines from the Janelia Gal4 collection, in which Gal4 is expressed under the control of cut enhancer fragments (Pfeiffer et al., 2008) (Figure 7—figure supplement 1A), we identified one line, R35B08-Gal4, which drove UAS-CD8-RFP expression specifically in future RPs at metamorphosis (Figure 7—figure supplement 1B)”. We also added one new figure panel (Figure 7—figure supplement 1B) demonstrating that CD8-RFP driven by this newly-identified cut enhancer-Gal4 colocalizes with endogenous Cut in future renal progenitors at the onset of metamorphosis.

Furthermore, to provide more detailed information regarding how we decided the size and sequence of the EMSA probes, we have now added the consensus TCF-binding site, helper site and Br-Z2-binding site to the figure legend of Figure 7—figure supplement 2.

3) Another substantial difficulty was that the inferences regarding developmental transitions that occur at the onset of metamorphosis. These were often difficult to follow and not written in a way that is easily grasped by non-experts. The same is true for the G-TRACE results. The authors run through these data fairly rapidly without making interim conclusions or putting their observations into context. There is a lot for the reader to keep track of cell name, cell position in the different organs, location of different organs and organ parts, ploidy, cut expression status, Esg status, developmental stage, and potentially unfamiliar acronyms build up quickly. Since this description sets up the remainder of the paper, it is important to rewrite these sections with a general reader in mind to help increase the appeal to the readership of eLife.

We thank the reviewers for this constructive suggestion. Since fly renal tubule is not a widely-used model system, its anatomy, cell composition and dynamic changes throughout developmental stages could cause potential difficulties for general readers. Therefore, to help increase the appeal of our manuscript to the broad readership of *eLife*, we have now carefully labelled the organ parts, such as midgut, hindgut, lower ureter, upper ureter and lower tubule, in most panels of main Figure 1 and 2.

As for the G-TRACE experiment, we have now added two new figure panels to show the genetic schema (Figure 3A) and experimental timeline (Figure 3B) of this lineage-tracing experiment. Furthermore, we included detailed description of the lineage-tracing strategy “We used temperature-sensitive esg-Gal4, UAS-CD8-GFP, tub-Gal80ts system to drive FLP (flippase) expression, transferred animals from permissive temperature (18°C) to restrictive temperature (29°C) at late 3rd instar larval stage and analyzed the renal phenotypes at early adult stages (Figure 3A,B). Upon esg-Gal4-driven expression of FLP recombinase, a transcriptional stop cassette flanked by FRT sites is excised, resulting in lacZ expression in Esg^+^ progenitors and all their subsequent daughter cells (lineage expression; Figure 3A). Meanwhile, GFP reveals the real-time expression of esg-Gal4 (Figure 3A)” (subsection “RPs displayed MP characteristics upon Cut depletion”). Furthermore, in light of reviewers’ comments, we have now added an interim conclusion “These results strongly suggested that, upon cut depletion, Esg^+^ progenitors migrate normally onto the renal tubules, yet fail to switch into renal identity and differentiate into midgut cells along renal tubules” (subsection “RPs displayed MP characteristics upon Cut depletion”).